

# Multi-decadal pacemaker simulations with an intermediate-complexity climate model

Franco Molteni [1,2], Fred Kucharski [1] and Riccardo Farneti [1]

[1] Abdus Salam International Centre for Theoretical Physics, Trieste, Italy

5    [2] European Centre for Medium-Range Weather Forecasts, Reading, United Kingdom

*Correspondence to*: Fred Kucharski (kucharsk@ictp.it)

**Abstract.** In this paper, we first describe the main features of a new version of the International Centre for Theoretical Physics global atmospheric model (SPEEDY) with improved simulation of surface fluxes, and the formulation of a 3-layer thermodynamic ocean model (TOM3) suitable to explore the coupled extratropical response to tropical ocean variability.

10    Then, we present results on the atmospheric model climatology, highlighting the impact of the modifications introduced in the model code, and show how important features of interdecadal and interannual variability are simulated in a "pacemaker" coupled ensemble of 70-year runs, where portions of the tropical Indo-Pacific are constrained to follow the observed variability.

Despite the very basic representation of variations in greenhouse forcing and heat transport to the deep ocean (below the 15    300m domain of the TOM3 model), the coupled ensemble reproduces the variations in surface temperature over land and sea with very good accuracy, confirming the role of the Indo-Pacific as a "pacemaker" for the natural fluctuations of global-mean surface temperatures found in earlier studies. Atmospheric zonal-mean temperature trends over 50 years are also realistically simulated in the extratropical lower troposphere and up to 100 hPa in the tropics.

On the interannual scale, SST variability in sub-tropical and tropical regions not affected by SST relaxation is 20    underestimated (mostly because of the absence of dynamically induced variability), while extratropical SST variability during the cold seasons is comparable to observed statistics. Atmospheric teleconnections patterns and their connections with SST are reproduced with high fidelity, although with local differences in the amplitude of regional features (such as a larger-than-observed response of extratropical SST to North Atlantic Oscillation variability). The SPEEDY-TOM3 model also reproduces the observed connection between averages of surface heat fluxes over the oceans and land surface air 25    temperature in the wintertime northern extratropics.

Overall, as in earlier versions of SPEEDY, the fidelity of the simulations (both in terms of climatological means and variability) is higher near the surface and in the lower troposphere, while the negative impacts of the coarse vertical resolution and simplified parametrizations are mostly felt in the stratosphere. However, the improved simulation of surface heat fluxes and their impact on extratropical SST variability in this model version (obtained at a very modest computational 30    cost) make the SPEEDY-TOM3 model a suitable tool to investigate the coupled response of the extratropical circulation to interannual and inter-decadal changes of tropical SST in ensemble experiments.



## 1 Introduction

Since its first release twenty years ago (Molteni 2003), the intermediate-complexity general circulation model (GCM) developed at the International Centre for Theoretical Physics (ICTP) has been used in several studies on atmospheric and
climate variability, both in its atmosphere-only version (see Kucharski et al. 2006, 2013 and references within) and coupled to both regional and global ocean models (Bracco et al. 2005, Kucharski et al. 2016). In its globally-coupled version, the model has also been used to develop prototypes of coupled data assimilation schemes (Sluka et al. 2016), and to estimate the large-scale impact of land surface modifications by human activities (Li et al. 2018). Atmospheric fields from simulations with prescribed sea-surface temperature (SST) have also been used to force global ocean models in studies of oceanic
decadal variability (Farneti et al. 2014). In the following, we will refer to the model using its acronym SPEEDY (for Simplified Parametrizations, primitivE-Equation DYnamics); an extensive list of publications where the SPEEDY model was used can be found on http://users.ictp.it/~kucharsk/speedy-doc.html .

One of the problems investigated with the atmosphere-only version of SPEEDY was the relationship between planetary-wave variability in the northern extratropics during boreal winter and the thermal forcing originated by surface heat fluxes
from the northern oceans (Molteni et al. 2011). Since such a forcing is actually dependent on the specific phase of the planetary waves with respect to the land-sea distribution, as originally postulated by theories of thermal equilibration (Mitchell and Derome 1983; Marshall and So 1990), the feedbacks between atmospheric circulation and surface heat flux variability can produce a regime behaviour in non-linear dynamical models (Molteni and Kucharski 2019). This variability also induces a change in total amount of heat transferred from the northern oceans to the extratropical atmosphere, and acts
as a source of natural fluctuations for surface air temperature (SAT) on continental scales (Molteni et al. 2017).

Decadal-scale variability in ocean-atmosphere heat exchanges was advocated as a major driver of the slowdown in near-surface global warming occurred at the beginning of the 21st century (Trenberth et al. 2014). By using a coupled GCM in a so-called "pacemaker" configuration, where ocean variables in a specific region are constrained to follow the observed variability, Kosaka and Xie (2013, 2016) investigated the role of decadal SST variability in eastern and central tropical
Pacific as a driver of global and regional SAT variability. In a subsequent analysis of those century-scale pacemaker simulations, Yang et al. (2020) argued that an increased natural variability in northern-hemisphere SAT was originated in recent decades by a 'synchronization' between forcings originated by tropical Pacific variability and anomalies in a specific planetary wave pattern over the northern extratropics (the Cold Ocean – Warm Land, or COWL pattern defined by Wallace et al. 1996). Although Yang et al. (2020) argued that such synchronization was just coincidental, since it was not found in
the record for earlier decades, one cannot rule out the possibility that interdecadal changes in tropical-extratropical teleconnections may have caused the COWL variability to go in phase with tropical Pacific SST in the most recent decades.

Due to the large amount of internal atmospheric variability in the northern extratropical circulation, reliable estimates of the impact of both tropical and extratropical thermal forcings require the production of ensemble simulations with many members (Kay et al. 2015; Maher et al. 2019) Although the reduced complexity of the SPEEDY atmospheric model allows



the production of multi-decadal ensembles with prescribed SST at a small computational cost (Bracco et al. 2004), such an advantage over more complex GCMs is significantly reduced when SPEEDY is coupled to a full-complexity ocean model (which takes most of the required computing resources). On the other hand, if the goal is to produce pacemaker simulations where a significant part of tropical SST variability is constrained to follow the observed variability, and the main focus of investigation are the extratropical teleconnections, one may wonder if a realistic simulation of the extratropical coupled

variability can also be achieved using a thermodynamical ocean model at a much reduced computational cost. This may be particularly relevant if one is specifically interested in ocean-atmosphere variability associated with significant surface heat flux variations. Indeed, intriguing results on the decadal variability of North Atlantic SST have been obtained with atmospheric GCMs coupled to a simple 'slab' ocean (e.g. Clement et al. 2015).

    This study presents results from multi-decadal simulations obtained with a new version of the SPEEDY model (named

version 42, v.42 in brief), either forced by prescribed SST or coupled to a 3-layer thermodynamic ocean model (referred to as TOM3) which can be constrained to reproduce the observed SST variability in selected regions. Since the coarse resolution of the SPEEDY versions used so far (with a horizontal grid of 96 x 48 points) prevents a detailed simulation of the pattern of surface heat fluxes in regions of complex coastlines or strong SST gradients, SPEEDY v.42 includes code to spatially interpolate near-surface atmospheric variables to a higher resolution grid, on which the evolution of land and ocean

conditions is either prescribed or evolved by coupled modules. For the ocean, the TOM3 model is run on a 1-degree gaussian grid, and simulates ocean temperature variability in the top 300 m of the ocean. As in all thermodynamic ocean models, a seasonally-varying forcing representing the convergence of heat flux due to ocean transport (often referred to as Q-flux) needs to be prescribed within TOM3 in order to maintain a realistic climatology.

    Section 2 of the paper provides a description of the main changes introduced in version 42 of SPEEDY, and describes the

geometry and formulation of the TOM3 model. Since the tuning of the new atmospheric code was performed by comparing the results of 30-year integrations with prescribed SST, the impact of the SPEEDY changes is illustrated in Section 3 using results from a 5-member ensemble forced by SST from the ERA5 re-analysis (Hersbach et al. 2020). Section 4 presents results from a second 5-member ensemble, performed by coupling SPEEDY to the TOM3 model in pacemaker mode, with SST variability constrained in those parts of the tropical Indo-Pacific known to be sources of important teleconnections. This

coupled ensemble covers the period 1950-2020, which includes the 'historical' period used in HighResMIP simulations (Haarsma et al. 2016). Diagnostics on interdecadal and interannual variability simulated by the coupled ensemble are discussed in Sect. 4, showing how this intermediate-complexity coupled model manages to reproduce a realistic relationship between variability of surface heat fluxes and continental SAT over the northern extratropics. Finally, results are summarised and discussed in Section 5, where plans for future developments of the coupled model and its use in specific research

projects are also presented.



## 2 Model formulation

### 2.1 New features of SPEEDY version 42

The modifications introduced in version 42 of SPEEDY (with respect to the version used in Kucharski et al. 2013) affect both the dynamics and the physical parametrizations of the model. First of all, while the default spectral truncation of the model has been kept at T30 (triangular truncation at zonal and total wavenumbers m=30, n=30), the gaussian grid on which non-linear terms are computed has been changed from the standard "quadratic" grid of 96x48 point to a "cubic" grid of 120x60 points, with resolution of 3 degrees. A "cubic" grid, which prevents aliasing of third-order non-linear terms, was introduced recently in the dynamical core of the ECMWF (European Centre for Medium-range Weather Forecasts) atmospheric model (Malardel et al. 2016), where it produced a more realistic energy distribution in waves close to the spectral truncation, also requiring a smaller amount of horizontal diffusion.

The second and most important change is in the interface between atmospheric and surface variables, and the computation of surface fluxes. In order to compensate for the coarse vertical resolution, SPEEDY does not use directly variables at the lowest model level to compute surface fluxes; instead, the model interpolates wind components, temperature and humidity from the two lowest model levels to the actual surface height to create near-surface air variables used in the flux computations. In v.42, all variables needed to define the near-surface values are also horizontally interpolated to a surface gaussian grid of higher resolution; for computational efficiency, the surface grid is defined as having 3 times the points of the atmospheric grid in both directions, and therefore has a default resolution of 1 degree.

Land and ocean variables are also prescribed (or evolved by a coupled module) on the surface grid, and surface fluxes are computed on the 1-degree grid from these variables and the near-surface air fields. When either a land or ocean model is coupled to SPEEDY, these fluxes are passed as input at the surface grid resolution; in order to compute physical tendencies for the atmospheric model, the fluxes are also interpolated back to the atmospheric cubic grid. Because of non-linearities involved in the computation of surface fluxes (e.g, their dependence on near-surface wind speed and stability), the fluxes returned to the atmospheric model differed from those derived from surface fields defined on the coarser atmospheric grid. This methodology also provides a more accurate input to coupled land and ocean/ice models, and allows a proper comparison with fluxes derived from re-analyses or complex climate models. The process is illustrated in Fig. 1 using mean model variables for January, showing the input to the SAT computation, the interpolated SAT, the observed SST and the turbulent heat flux computed on the surface grid, and finally the heat flux interpolated back to the atmospheric grid. More detailed results on surface fluxes are presented in Sect. 3 below.

With regard to parametrizations, apart from minor changes in the parameters of the surface flux bulk formulae, main changes have been introduced in the formulation of radiative processes and both the horizontal and vertical diffusion of moisture.

For shortwave radiation, modifications are as follows:



- The top-of-the-atmosphere solar input can now be specified either as a daily-mean field (the only option available in the standard version 41) or a field evolving through a daily cycle (a feature introduced in version 41.5 and used to produce estimates of daily maximum/minimum temperature, as in Gore et al. 2019).
- A revised ozone climatology (defined by simple functions of seasonal time and latitude, tuned by comparison with ERA5 ozone data for the recent decades) has been introduced, covering the top three (instead of two) levels of the model, and therefore accounting for the lower tropopause height in high-latitude regions.
- The albedo of non-stratiform clouds depends on solar zenith angle, and is therefore higher at high latitudes.

For longwave radiation:

- The increase in absorptivity due to clouds in the two water-vapour bands has been revised, and it is now height dependent.
- The empirical correction terms active in the top two levels (to compensate for the absence or poor representation of ozone and water vapour long-wave cooling in the stratosphere) have been modified and reduced, as a result of a better specification of ozone-induced warming.

Modifications to moisture diffusion are as follows:

- Vertical diffusion of specific humidity is now limited to the lowest 3 layers (below $\sigma = p/p_s \approx 0.7$) in cloud-free areas, to the cloud-top level otherwise (instead of being applied everywhere below $\sigma = p/p_s \approx 0.5$).
- In order to avoid spurious diffusion on mountain slopes (a consequence of adopting a $\sigma = p/p_s$ vertical coordinate), horizontal diffusion of humidity acts on the deviation of the specific humidity field from a reference, topography-dependent state. While in earlier versions such a reference state was computed in spectral space using a standard vertical profile, in v.42 the reference state is computed in grid-point space by horizontally smoothing the relative humidity field and multiplying it by the saturation specific humidity corresponding to the actual temperature. In this way, local topographic gradients are represented much more accurately.

The impact of the changes described above on the model climatology will be shown and discussed in Sect. 3. In terms of computing costs, version 42 requires about 25% more computing time than the previous versions, but still runs very efficiently on the latest generation of workstations: when the model is allocated 4 cores and 8 GB of memory, one year of simulation requires only 6.5 to 7 minutes (depending on the specific processor).

## 2.2 The TOM3 thermodynamic ocean model

TOM3 is a 3-layer thermodynamic ocean model, which replaces a single-layer slab model as the default, single-executable option for ocean coupling in SPEEDY v.42. The model computes the evolution of sea water and sea ice temperature in the top 300 metres of the oceans, with a minimum imposed depth of 90 m. In the sub-sections below, all variables are defined at each ocean grid point, so their dependence on horizontal coordinates is omitted.





**Figure 1:** Interpolations involved in the computation of surface fluxes in SPEEDY v.42, illustrated using January-mean temperature and heat fluxes from an integration with observed SST. The temperature on the lowest two atmospheric models (panels a and b) are vertically and horizontally interpolated to produce a near-surface air temperature on the 1-degree surface grid (c). The same process is applied to compute near-surface wind and humidity. Using these fields and the surface temperature (d) defined on the surface grid, turbulent (sensible + latent) heat fluxes are computed (e), and passed as input to coupled ocean and land models. Turbulent heat fluxes are then interpolated back to the atmosphere grid (f) to produce tendencies for the atmospheric model.



### 2.2.1 TOM3 geometry and variable definitions

If $D^*$ is the actual ocean depth at any model grid point, the total depth $D$ covered by the TOM3 model is:

$$D = max\ (90m,\ min\ (300m,\ D^*))\tag{1a}$$

This domain is divided into three layers, the top two representing a mixed layer of total depth $D_{ml}$ :

$$D_{ml} = min\ (D/3,\ D'(\varphi))\tag{1b}$$

In the experiment described in this paper, $D'(\varphi)$ is a simple function of latitude ranging from 30m at the equators to 60m at the poles. Maps of $D$ and $D_{ml}$ are shown in Fig. S1 of Supplementary Information. The three individual layers have depths given (from top to bottom) by:

$$D_1 = 10m,\quad D_2 = D_{ml} - D_1\ ,\quad D_3 = D - D_{ml}\tag{1c}$$

and mass per unit surface given by:

$$M_k = \rho_w\ D_k,\quad k = 1,\ ..,\ 3\tag{1d}$$

where $\rho_w$ is a reference density of sea water. Furthermore, the mass $M_1$ of the top layer is divided into a sea-ice fraction $f_i\ M_1$ and a water fraction $(1 - f_i)\ M_1$ . The fraction of ice mass $f_i$ can be expressed as the product of the surface concentration of ice $s_i$ and an equivalent mass depth $d_i$ :

$$f_i =\ s_i\ d_i\tag{2a}$$

where $d_i$ is related to the actual ice thickness $\theta_i$ by:

$$d_i = \theta_i\ \rho_i\ /\ D_1\ \rho_w\tag{2b}$$

and $\rho_i$ is a reference density of sea ice ( $\rho_i \approx 0.9\ \rho_w$). In the following, we will refer to the ice fraction of layer 1 as to the sea-ice layer.

The TOM3 model can be run in either a prescribed-ice mode (mode 1) or in an interactive-ice mode (mode 2). In mode 1, the evolution of the following prognostic variables is computed:

- $T_1^w$, $T_2^w$, $T_3^w$ : temperature of sea water in the three layers;
- $T_m^i$ : mean temperature of the sea-ice layer.

while $s_i$ and $d_i$ are prescribed from observed or climatological values. In mode 2, $s_i$ and $d_i$ become additional prognostic variables. In both modes, two additional ice temperatures are prescribed or defined diagnostically:

- $T_0$ : temperature at the bottom of the sea-ice layer, assumed to be equal to the freezing temperature of sea water;
- $T_1^i$ : temperature in the near-surface part of the ice layer, used by the atmospheric model to compute skin temperature and surface heat fluxes.





If we indicate the temperature of 0 °C as $T_{0C}$, $T_1{}^i$ is defined by fitting a parabolic profile $T^i(z)$ to the temperature within the
ice layer, consistent with the mean value $T_m{}^i$, the lower-boundary value $T_0$ and an upper-boundary value $T_u{}^i$ given by the
empirical relationship:

$$T_u{}^i - T_{0C} = \gamma_i \ ( \ T_m{}^i - T_0 \ ) \tag{3}$$

where $\gamma_i$ ranges between 2 and 2.5 depending on whether the surface heat fluxes are warming or cooling the ice (more details
are given in the Appendix).

All experiments described in Sect. 4 have been run in mode 1; tuning of additional parameters specific for mode-2
integrations is under way, and results will be presented in a subsequent paper.

### 2.2.2 Heat content and sub-surface heat fluxes

From the variables defined above, the heat content (per unit surface) of sea water and sea ice in the three layers is defined
as the difference from a reference state where all sea water is at freezing temperature $T_0$ :

$$HC_1{}^w = M_1 \ (1 - f_i) \ c_w \ (T_1{}^w - T_0) \tag{4a}$$
$$HC_1{}^i = M_1 \ f_i \ [ \ c_i \ (T_m{}^i - T_0) - L_f \ ] \tag{4b}$$
$$HC_2 = M_2 \ c_w \ (T_2{}^w - T_0) \tag{4c}$$
$$HC_3 = M_3 \ c_w \ (T_3{}^w - T_0) \tag{4d}$$

where $c_w$, $c_i$ and $L_f$ are respectively the specific heat for sea water and sea ice and the latent heat of fusion. To compute the
evolution of the heat content, heat fluxes at the top and bottom of each layer are needed. The surface heat fluxes, computed
by the atmospheric model, are:

- $F_s{}^w$ : net solar heat flux over sea water
- $F_{ns}{}^w$ : net non-solar (sensible + latent + net longwave radiation) heat flux over sea water
- $F_s{}^i$ : net solar heat flux over sea ice
- $F_{ns}{}^i$ : net non-solar (sensible + latent + net longwave radiation) heat flux over sea ice

In addition, TOM3 computes the heat fluxes at the boundaries between the water layers:

$$F_1{}^w = 2 \ k_1 \ ( \ T_1{}^w - T_2{}^w ) \ / \ ( D_1 + D_2 ) \tag{5a}$$
$$F_2{}^w = 2 \ k_2 \ ( \ T_2{}^w - T_3{}^w ) \ / \ ( D_2 + D_3 ) \tag{5b}$$

and the flux at the lower boundary of the sea ice, computed from the derivative of the parabolic temperature profile $T^i(z)$ at
the ice bottom:

$$F_0{}^i = - \ k_i \ \partial T^i(z)/\partial z \quad at \ z = \theta_i \tag{5c}$$

with all fluxes defined as positive downward. In Eq. 5c, $k_i$ is the thermal conductivity of ice. In Eq. 5a and 5b, the
coefficients $k_1$ and $k_2$ assume different values according to the sign of the vertical temperature gradient, being smaller/larger



when temperature decreases/increases with depth to account for convective mixing. Furthermore, where sea-ice is present
and $T_1{}^w$ is lower than both $T_2{}^w$ and $T_{0C}$ , an additional (negative) term $F_c{}^w$ is added to both $F_1{}^w$ and $F_2{}^w$ to account for
convection bringing cold water from the top to the bottom water layer.

A schematic illustration of the model variables and fluxes is given in Fig. 2.

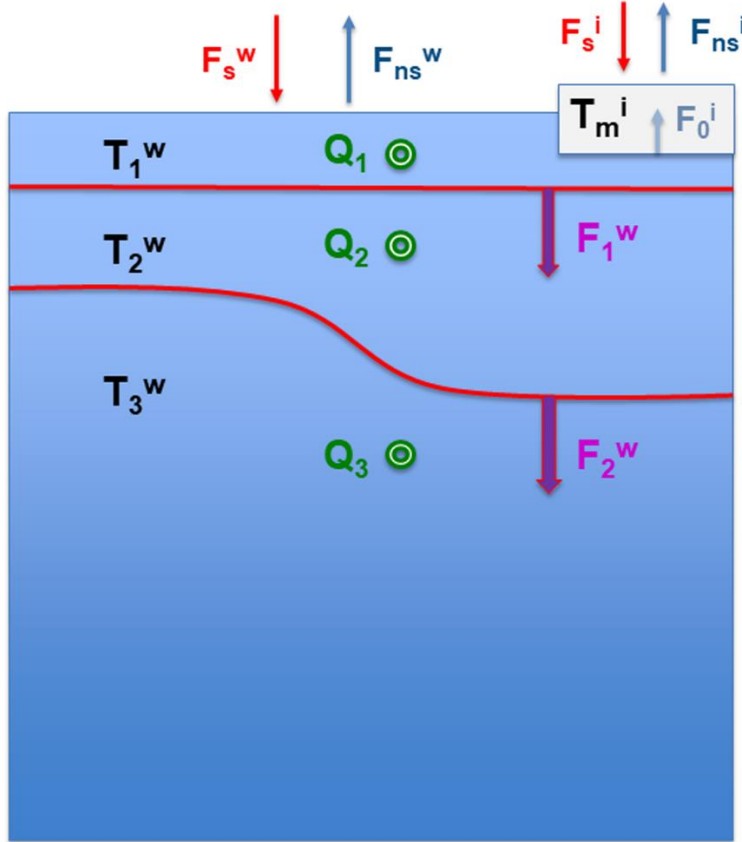


**Figure 2:** Prognostic variables (in black) and heat fluxes in TOM3. Red and blue arrows at the ocean surface indicate solar (red) and non-solar (blue) heat fluxes at the water and ice surface. The purple arrows with red outline indicate heat fluxes between the three water layers, and the blue-grey arrow represents the heat flux between water and ice in layer 1. The green dots indicate the Q terms, representing the climatological convergence of the dynamical heat transport (possibly modified by regional relaxation to observed SST). See text in Sect.
2.2 for further explanations.



### 2.2.2 Prognostic equations for sea water and sea ice temperature

With all heat fluxes defined as in the previous sub-section, the time evolution of the heat content is determined by the convergence of net heat flux into each layer, leading to prognostic equations for sea water and ice temperature. For sea water, the equations read:

$$[ c_w (1 - f_i) M_1 ] \partial T_1^w / \partial t = (1 - s_i) ( F_s^w + F_{ns}^w ) + s_i F_0^i - F_1^w + Q_1 \tag{6a}$$

$$( c_w M_2 ) \partial T_2^w / \partial t = F_1^w - F_2^w + Q_2 \tag{6b}$$

$$( c_w M_3 ) \partial T_3^w / \partial t = F_2^w + Q_3 \tag{6c}$$

In Eq. 6a-c, the terms $Q_1$, $Q_2$ and $Q_3$ represent the climatological, seasonally-varying convergence of heat from dynamical transport. In brief, they are computed by first running the model driven by near-surface air and ice variables from the ERA5 re-analysis (Hersbach et al. 2020), and with the $Q$ terms set to zero. The average monthly tendencies of heat content produced in this way are compared with tendencies from the World Ocean Atlas (WOA09; Locarnini et al., 2010) ocean climatology: the Q terms are defined by the differences between the WOA09 and model tendencies.

For sea ice, the time derivative of the heat content is given by:

$$\partial HC_1^i / \partial t = s_i ( F_s^i + F_{ns}^i ) - s_i F_0^i \tag{7a}$$

However, the model assumes that only a fraction $\alpha_i$ of the heat entering the ice layer is converted in temperature change, with the remaining fraction changing the ice mass. Specifically, TOM3 assumes that $f_i$ increases when sea ice is cooling and decreases when sea ice is warming. Therefore the ice temperature tendency is reduced, in absolute value, with respect to schemes which assume a constant ice thickness (one should note that, in order to obtain realistic sea ice temperature variations, such constant sea ice thickness is often set to a large value, which is unlikely to be appropriate for small concentration values; for example, in ERA5 the sea ice thickness is set to 1.5 m). With this assumption, the prognostic equation for sea ice mean temperature is given by:

$$( c_i f_i M_1 ) \partial T_m^i / \partial t = \alpha_i \partial HC_1^i / \partial t \tag{7b}$$

Starting from values of $HC_1^i$ and $T_m^i$ at time t, Eqs. 7a and 7b are used to compute the corresponding values at time t+δt; then, the mass fraction of sea ice $f_i$ is redefined to satisfy the heat content conservation:

$$HC_1^i (t+\delta t) = M_1 f_i (t+\delta t) [ c_i (T_m^i (t+\delta t) - T_0) - L_f ] \tag{8}$$

with a corresponding adjustment taking place in the sea water temperature of layer 1.

Finally, the model creates new sea ice if $T_1^w$ becomes lower than $T_0$, melts sea ice if $T_m^i$ becomes greater than (or close enough to) $T_0$. Since the coupled experiments discussed here are run in mode 1, details about the rules used for such phase transitions are omitted here; it suffices to say that any such transformation conserves the total heat content of layer 1, so that the evolution of $HC_1$ is only determined by the fluxes at the layer boundary and the $Q_1$ flux:



$$\partial\,(\,HC_1{}^w + HC_1{}^i\,)\,/\partial t = (1 - s_i)\,(\,F_s{}^w + F_{ns}{}^w\,) + s_i\,(\,F_s{}^i + F_{ns}{}^i\,) - F_1{}^w + Q_1 \tag{9}$$

If the sea-ice is run in (non-interactive) mode 1, the updated value of $f_i$ is actually replaced using new prescribed values of ice concentration and thickness at the following time step. In interactive mode 2, an empirical relationship is used to compute

values of $s_i$ and $d_i$ at time t+δt from the updated value of $f_i$ (see Appendix).

### 2.2.4 Regional relaxation towards observed SST

Pacemaker experiments, such as those performed by Kosaka and Xie (2016), require the relaxation of upper-ocean variables towards observed, time-evolving values. In TOM3, a relaxation of near-surface sea water temperature towards observed SST can be activated in a domain specified by spatial masks. In any day and at any grid point $j$, a relaxation heat flux is defined

as:

$$F_{rel}\,(j) = k_{rel}\,[\,SST_{obs}\,(j) - T_1{}^w\,(j)\,]\,R\,(j) \tag{10}$$

where $R(j)$ is the local value of the relaxation mask $R$ (varying between 0 and 1). The area-weighted average of $F_{rel}\,(j)$ over all grid points, $F^*{}_{rel}$, is then computed at each step. If this is different from zero, the relaxation flux would produce a spurious source or sink of energy for the global ocean. In TOM3, since the relaxation is supposed to be applied in tropical

regions where tendencies are mainly driven by ocean transport, we assume that changes in mixed-layer heat content induced by the relaxation should implicitly be seen as the results of changes in equatorial upwelling and sub-tropical cell motion. Therefore, we impose that the global change in mixed-layer heat content be compensated by a change in the heat content of the bottom layer, such that the global integral of the two terms is zero.

This is achieved by defining a so-called compensation mask $R_c\,(j)$ (also varying between 0 and 1), covering the region of

the sub-tropics where the compensating transport is supposed to take place. If we indicate with $R_c^*$ the area-weighted global average of $R_c\,(j)$, the relaxation process acts by modifying the climatological transport term $Q$ in Eqs. 6a-c as follows:

$$Q'_1\,(j) = Q_1\,(j) + (1/3)\,F_{rel}\,(j) \tag{11a}$$
$$Q'_2\,(j) = Q_2\,(j) + (2/3)\,F_{rel}\,(j) \tag{11b}$$
$$Q'_3\,(j) = Q_3\,(j) - (F^*{}_{rel}/R^*{}_c)\,R_c\,(j) \tag{11c}$$

For the pacemaker experiments presented in Sect. 4, the relaxation mask has non-zero values over the central-eastern tropical Pacific and the western Indian Ocean. These regions are chosen because they are important sources of atmospheric teleconnections, characterised by a positive correlation between SST and rainfall anomalies (see e.g. Fig. 1 in Molteni et al. 2015). However, heat fluxes play a different role in the SST variability of two regions: in the central/eastern Pacific, they tend to dissipate the SST anomalies induced by ocean dynamics, while in the Indian ocean they may have a reinforcing role.

As a consequence, a stronger relaxation is needed in the former region to constrain SST variability, and therefore the relaxation mask (shown in the upper panel of Fig. 3) has been set to larger values in the Pacific than in the Indian sector. The compensation mask in our experiments, also shown in Fig. 3, covers the whole sub-tropical region in the Indo-Pacific Ocean.




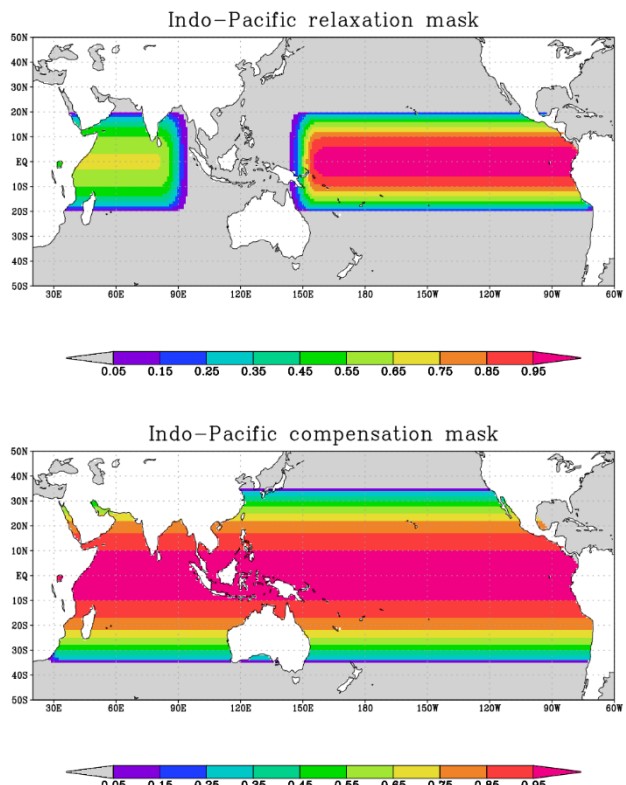

**Figure 3:** Relaxation mask (top) and compensation mask (bottom) used in the pacemaker experiments described in Sect. 4. See Sect. 2.2.4 for further details.

## 3 Impact of atmospheric model changes in simulations with prescribed SST

In this section, we discuss the impact of the changes to the atmospheric model formulation (described in Sect. 2.1) on the model mean state and its variability. Results come from a 5-member ensemble of runs in which SST and sea-ice concentration from ERA5 are prescribed as boundary conditions. The ensemble members cover the period January 1981 – December 2020, and are initialised from different initial states. Statistics on the annual and seasonal means are presented for the period 1981-2010, which is used as a reference period for the assessment of the SPEEDY climatology because of the availability of multiple observational datasets and state-of-the-art GCM simulations used for comparison. Since the estimation of second-order moments is affected by a higher uncertainty, teleconnection patterns (defined as covariances with selected indices) are estimated from data covering the full 40-year integrations.



## 3.1 Surface heat fluxes

Since the main objective of the changes to the model formulation is to produce a more realistic simulation of surface fluxes, we first assess how the annual-mean patterns of such fluxes are simulated and to what extent they achieve a global balance
consistent with current estimates of net surface warming. Since re-analyses run with prescribed SST (as ERA5) are not expected to achieve a closed energy balance at the surface (see Hersbach et al. 2020), we prefer to compare the annual-mean fluxes from SPEEDY with those from an ensemble of prescribed-SST simulations performed with the ECMWF model as a contribution to the EU-funded PRIMAVERA project (Roberts et al. 2018), following the HighResMIP protocol of CMIP6 (Haarsma et al. 2016) and covering the historical period 1950-2014. The version of the ECMWF model used for this
ensemble is close to the version used in ERA5, and includes some minor parameter adjustments to achieve a realistic surface energy balance.

Fig. 4 presents the balance of the solar (net short-wave radiation) and non-solar (net long-wave radiation, sensible and latent heat fluxes) components of the annual-mean surface heat fluxes; in Fig. 5, maps for the three components of the non-solar flux are shown separately. Global averages of these fluxes are listed above each panel, and also reported in Table 1,
where they are compared with estimates from Wild et al. (2015) based on a combination of observations and CMIP5 model data. In addition, zonal means of surface fluxes from the SPEEDY and ECMWF ensembles, and of their mean absolute differences, are shown in the Supplementary Information figure S2.

The spatial patterns of solar and non-solar fluxes from SPEEDY, shown in Fig. 4, compare quite well with the ECMWF counterparts. The SPEEDY net solar radiation is lower than the ECMWF flux over the subtropical oceans, higher in the
extratropics; the SPEEDY global average is just outside the range estimated by Wild et al. (2015), but still quite realistic. A positive difference of about 6 W/m$^2$ with respect to the ECMWF model is also found in the global average of the non-solar heat flux, resulting in a close match between the net global averages: 0.5 W/m$^2$ for SPEEDY and 0.7 W/m$^2$ for the ECMWF model, both values being well within the range estimated by Wild et al. (2015). Also, the maps for the net surface radiation in the two models (bottom panels in Fig. 4) show a good correspondence between ocean regions gaining or losing energy, an
important requisite if users want to couple SPEEDY to an ocean model.

When looking at the individual components of the non-solar heat flux (in Fig. 5), the net longwave radiation emerges as the fields with the larger discrepancies between SPEEDY and the ECMWF model, with the former showing larger values over most of the oceans. Among the various components of the heat fluxes produced by SPEEDY, this is the only one which shows a global average significantly outside the range estimated by Wild et al. (2015). Conversely, the patterns and global
averages of sensible and latent heat flux show a good agreement between the two models: the strong fluxes associated with western boundary currents are clearly visible in both models. The global averages from SPEEDY are well within the uncertainty range in Wild et al. (2015), and the lower average of latent heat flux with respect to the ECMWF model partially compensates the excess in net upward longwave radiation.





Since the upward flux of longwave radiation over the oceans can be reproduced with little uncertainty, the errors in the net longwave radiation over the ocean are bound to come from an insufficient downward emission from the SPEEDY atmosphere. To put this error in a proper context, it should be noted that the difference of 12 W/m$^2$ between SPEEDY and the ECMWF model corresponds to less than 4% of the average downward flux of longwave radiation at the surface.


|  | SPEEDY v.42 | ECMWF-Ah | Wild et al. (2015) |
|---|---|---|---|
| Net solar radiation (downw.) | 167.8 | 162.1 | 160 (154 / 166) |
| Net longwave radiation (upw.) | 69.8 | 58.0 | 56 (48 / 62) |
| Sensible heat flux (upw.) | 17.3 | 17.6 | 21 (15 / 25) |
| Latent heat flux (upw.) | 80.3 | 85.8 | 82 (70 / 85) |
| Net surface heat flux (downw.) | 0.48 | 0.74 | 0.6 (0.2 / 1.0) |

**Table 1:** Global averages of surface heat fluxes (for the 1981-2010 period) in multidecadal ensemble simulations with SPEEDY v.42 and the ECMWF atmospheric model (ECMWF-Ah; Roberts et al. 2018), compared with mean estimates and uncertainty ranges from Wild
et al. (2015). All data in W/m$^2$.



**Figure 4:** Left column: annual-mean climatology (for 1981-2010) of solar (top), non-solar (centre), and net surface heat fluxes (bottom), in W/m², simulated by the SPEEDY ensemble with prescribed SST (ens.653). Solar and net fluxes are downward, non-solar flux is upward. Right column: as in left column, but from an ensemble of 65-yr simulations with prescribed SST run at ECMWF (ECMWF-Ah; Roberts et al. 2018).



**Figure 5:** Left column: annual-mean climatology (for 1981-2010) of net surface long-wave radiation (top), sensible heat (centre), and latent heat fluxes (bottom), in W/m$^2$, simulated by the SPEEDY ensemble with prescribed SST (ens.653). All fluxes are upward. Right column: as in left column, but from an ensemble of 65-yr simulations with prescribed SST run at ECMWF (ECMWF-Ah; Roberts et al. 2018).






## 3.2 Atmospheric mean state and teleconnections

We now look at the impact of the model changes listed in Sect. 2.1 on the atmospheric mean state and variability. Since we want to focus on the changes to physical parametrizations, in Figures 6 to 8 we compare the mean state in the 5-member ensemble with prescribed SST with the state achieved in integrations performed with the same grid setting as in version 42, but no change in parametrizations with respect to the previous version.

Fig. 6 illustrates the impact of changes in solar and longwave radiation (including the new ozone climatology) on the bias of atmospheric temperature in the lower troposphere (at 100 hPa) and the upper troposphere (at 300 hPa). The Hovmoller diagrams show the biases in the northern hemisphere across the seasonal cycle for the two model configurations, and their difference. In the tropics, the warm biases have been reduced by about a factor of 2, with maximum values going from 8 to 4 degrees (similar changes are seen in the southern-hemisphere tropics, not shown). In high-latitude regions, a strongly seasonally varying bias (cold in winter-spring and warm in summer) was present in the lower stratosphere because of a too simplified prescription of the ozone climatology in the previous SPEEDY version. Changes to the ozone climatology and a consequent reduction of empirical correction terms have produced a more seasonally uniform cold bias in the northern polar region, with an amplitude comparable with the biases of state-of-the art models in this part of the atmosphere.

These changes have produced a strong reduction in the lower-stratosphere horizontal temperature gradient between the northern sub-tropics and the polar region (see Fig. 6e); this is bound to produce a reduction in the strength of the stratospheric polar vortex. We will comment on the implications of this reduction for the tropospheric flow in the North Atlantic region in the paragraphs below.

Fig. 7 shows the impact of the changes in the horizontal and vertical diffusion of moisture on the climatology of rainfall and 850 hPa wind during the Asian monsoon season. Before such changes, a too-weak correction of diffusion along orographic slopes produced a split between rainfall in the northern and central part of the Indian subcontinent (as well as in the adjacent seas), leaving the Ganges valley unrealistically dry (Fig. 7c). As a result, instead of the observed bimodal distribution with ocean and continental maxima, the rainfall climatology over South Asia showed a tri-polar structure with a strong maximum around 10ºN. This bias has been largely corrected by the new formulation of horizontal diffusion for humidity (see Fig. 7a), with smaller contributions also coming from the changes in vertical diffusion and long-wave absorptivity by clouds. The 850 hPa wind climatology has also been affected, with a further northward extension of westerly winds into the Indian sector (fig 7b). Unfortunately, the representation of the East Asian monsoon shows only minor improvements, and the simulation of rainfall connected to the Meiyu–Baiu front is still clearly deficient.

While the main focus in the development of v.42 has been on an improved representation of surface fluxes, earlier versions of SPEEDY were primarily tuned to produce a realistic simulation of the northern hemisphere (NH) circulation during the boreal cold season, and of the main teleconnection patterns which characterise its variability. It is therefore appropriate to verify whether v.42 has maintained a good fidelity in the simulation of the boreal winter circulation.





**Figure 6:** Bias of temperature (in °K) at 100 hPa (left) and 300 hPa (right) in SPEEDY ensembles with prescribed SST, computed against ERA5 data, for DJF 1981-2010. **a)** and **b)**: ensemble with v.42 (ens.653). **c)** and **d)**: ensemble with v.42 dynamics but v.41 parametrizations (ens.503). **e)** and **f)**: difference between the two ensembles, showing the impact of v.42 parametrizations.





**Figure 7:** Asian monsoon rainfall (left, in mm/day) and 850-hPa zonal wind (right, in m/s) in SPEEDY ensembles with prescribed SST, for JJA 1981-2010. **a)** and **b)**: ensemble with v.42 (ens.653). **c)** and **d)**: ensemble with v.42 dynamics but v.41 parametrizations (ens.503). **e)**: rainfall climatology from GPCPv2.3 (Adler et al. 2003). **f)**: zonal wind climatology from ERA5.





Fig. 8 shows the mean field of NH 500 hPa geopotential height in the December-to-February (DJF) season as simulated by the v.42 ensemble; both the full field (fig. 8a) and the deviation from its zonal mean (8b) are shown, and compared with the same fields from ERA5 data. Panels 8c and 8f show the difference of the v.42 height climatology from those computed respectively from ERA5 data and from simulations without parametrization changes.

Overall, SPEEDY v.42 has maintained a realistic representation of the NH wintertime climatology, although with some underestimation of the stationary wave amplitude in the Atlantic sector; the most evident discrepancy is found in the depth of the stationary trough over eastern North America and the Labrador Sea. The mean bias with respect to ERA5, shown in Fig. 8c, reaches a maximum amplitude of about 80 m over small portions of the northern oceans, but over most of the extratropical NH the amplitude of the bias is less than 40 m. Although the best state-of-the-art GCMs show smaller biases in the boreal winter, mid-tropospheric circulation, the amplitude of the v.42 bias is comparable to those of many models used for CMIP5 historical runs (see fig. 1 in Pithan et al. 2016).

However, comparing the 500-hPa height bias with the impact of the parametrization changes (Fig. 8f), it is evident that the two fields are positively correlated, particularly over Canada and the North Atlantic, where the parametrization changes have produced a dipole pattern projecting on the negative phase of the North Atlantic Oscillation (NAO). Since, in the real atmosphere, a connection between a stronger/weaker stratospheric polar vortex and a positive/negative phase of the NAO is well documented (Kidston et al. 2015; Charlton-Perez et al. 2018 and references within), the shift towards a negative NAO phase would appear to be consistent with the changes in polar stratospheric temperature shown in Fig. 6.

We further explore this issue in the context of a more general assessment of NH extratropical teleconnection patterns. For this purpose, and guided by many earlier studies, we define three teleconnection indices from the difference of anomalies in selected regions of the NH:

● North Atlantic Oscillation (NAO) index: mean-sea-level pressure anomaly, (35W-15E, 35N-50N) – (50W-0E, 55N-75N)

● Pacific – North American (PNA) index: 500-hPa height anomaly, 1/3 (180W-140W, 15N-25N) - 2/3 (180W-140W, 35N-55N) + 2/3 (125W-85W, 45N-65N) - 1/3 (95W-65W, 25N-40N)

● Stratospheric polar temperature (SPT) index: 100 hPa temperature anomaly, NH extratropics (25N-90N) – polar region (60N-90N)

The definition of the SPT index is such that a positive value is associated with an intensification of the lower stratosphere polar vortex; also, measuring the deficit of the polar temperature with respect to the whole northern extratropics, we prevent the index variability to be strongly dominated by the overall stratospheric cooling trend induced by the $CO_2$ increase.

Although each of the three indices is defined from a different anomaly field, to compare the associated variability in the mid-tropospheric circulation we show in Fig. 9 the covariances of the 500 hPa field with the normalised time series of the three indices: these can be interpreted as the 500-hPa anomaly associated with one positive standard deviation of each index. For consistency with the tropical teleconnections discussed in Sect. 4, which display a significant sub-seasonal variability (Ayarzaguena et al 2018; King et al. 2021), we compute these covariances from bi-monthly mean anomalies in January and



February (JF) over the 1981-2020 period. The same teleconnection patterns, computed from ERA5 data, are also shown for comparison.

For the NAO and PNA teleconnections, the SPEEDY anomaly patterns compare well with the ERA5 counterparts, although the amplitude of the PNA anomaly is about 25% smaller in the SPEEDY pattern than in ERA5 over the North Pacific, and the anomalies do not extend as far into the North Atlantic and Europe. For the NAO pattern, the overall amplitude is quite close between model and re-analysis, but again the SPEEDY anomalies do not extend as far eastwards as the ERA5 anomalies.

The height anomaly associated with SPT variability, on the other hand, has a significantly smaller amplitude in SPEEDY than in ERA5, although the projection on the positive NAO phase is correctly (and very clearly) reproduced by the model. However, when we look at the standard deviation of the SPT index in SPEEDY and ERA5, we find that a value of 1.4 °K for the model and 2.9 °K for ERA5. Therefore, while the stratospheric variability in SPEEDY is much lower than in reality (as may be expected because of the very coarse vertical resolution), the 500 hPa anomaly induced by the *same* stratospheric temperature variation is actually comparable with the re-analysis. This is shown more clearly in Fig. S3 of Supplementary Information, where the regressions of 500-hPa height against the *dimensional* SPT index are shown in units of m/°K. Using the SPEEDY regression to estimate the linear contribution of the changes in the lower-stratosphere climatology, Fig. S2 confirms the hypothesis that the shift towards a negative NAO induced by parametrization changes in v.42 (shown in Fig. 8f) is caused to a large extent by the reduction of the polar stratospheric cold bias (see Fig. 6).

Overall, we find no deterioration in the simulation of extratropical teleconnections in the latest version of SPEEDY, despite the increase in the mid-tropospheric geopotential bias in the northern extratropics during the boreal winter. The fact that changes which were beneficial for the reduction of stratospheric biases turned out to have a negative effect on the 500-hPa model climatology is most likely an indication of the presence of compensating errors in the representation of stratospheric and tropospheric processes. Interestingly, Pithan et al. (2016) showed that an underestimation of the stationary wave amplitude over North America and the North Atlantic, quite similar to that found in the SPEEDY v.42 climatology (see fig. 8d) and many CMIP5 models, could be reproduced by decreasing the strength of the orographic low-level drag in simulations performed with the UK Met Office Unified Model. Since currently the enhancement of surface drag over topography is rather crudely parameterized in SPEEDY, these results point to a direction for possible improvements which will be explored in future experimentation.







**Figure 8:** Left column: climatology of 500-hPa geopotential height for DJF 1981-2010 (in dam), from: **a)** the ensemble with prescribed SST (ens.653), and **d)** ERA5 data. Middle column (**b, e**): as on left column, but for the stationary waves at 500 hPa (i.e differences from the zonal-mean climatology). **c)**: bias of the 500-hPa height ensemble climatology with respect to ERA5. **f)**: difference with respect to an ensemble with v.42 dynamics but v.41 parametrizations (ens.503).



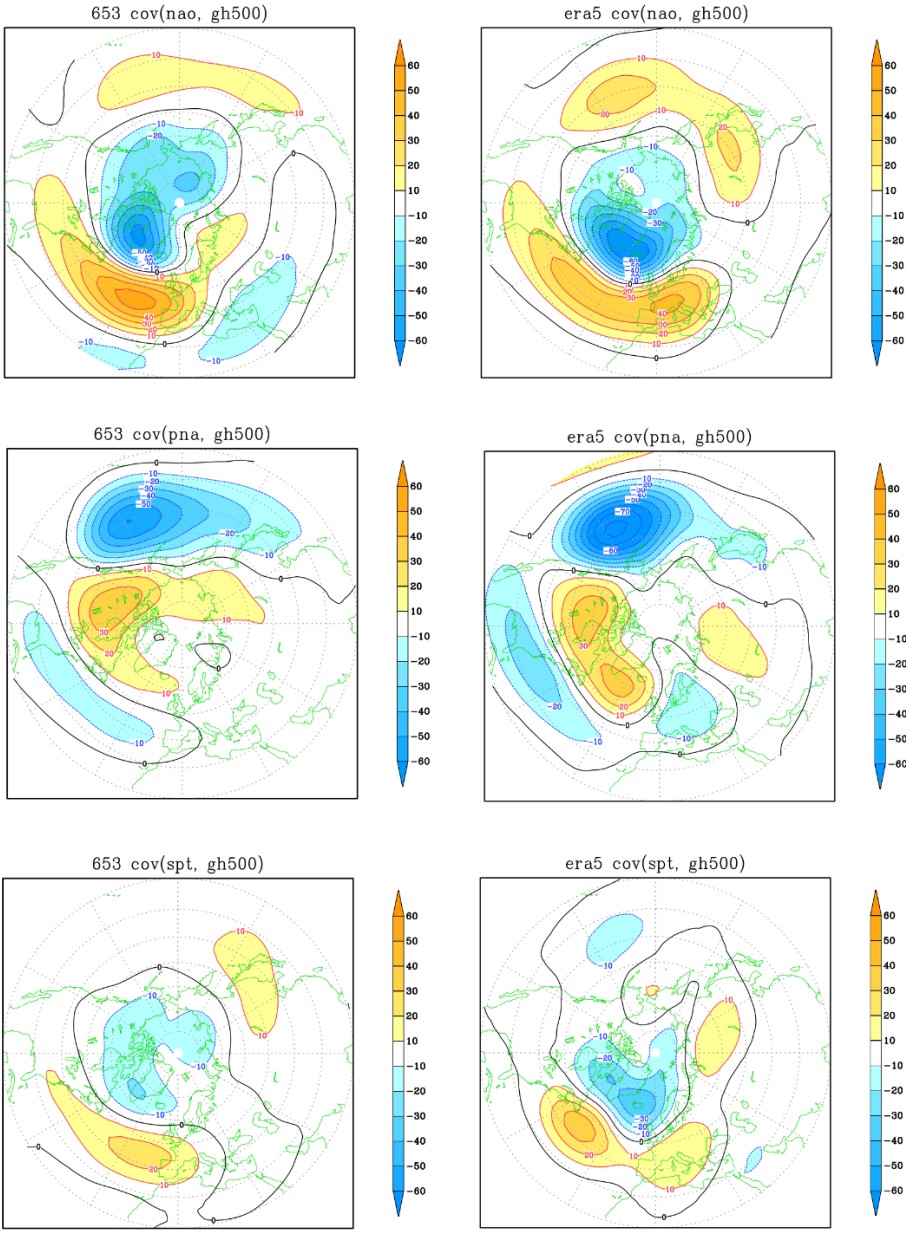

**Figure 9:** Covariance of JF 500-hPa height (in m) with the NAO (top), PNA (centre) and SPT (bottom) indices, for the ensemble with prescribed SST (ens.653, left) and ERA5 data (right) in JF 1981-2020.



## 4 Ocean and atmosphere variability in coupled pacemaker experiments

We now present results from a second 5-member ensemble, run with SPEEDY v.42 coupled to the TOM3 model, with
relaxation to observed SST activated in parts of the tropical Indo-Pacific ocean (as explained in Sect. 2.4). The five ensemble
members run from January 1950 to December 2020, and are initialised from slightly different atmospheric initial conditions.
The initial condition for the TOM3 model is, for all members, a climatological field appropriate for the decade started in
1950; this is obtained as a modification of the WOA09 climatology based on the ERA5 SST difference between different
periods. As stated in Sect. 2.2, these integrations have been run with prescribed ice concentration and thickness, while
temperature of both sea water and sea ice are evolved interactively by TOM3 according to Eqs. 6 and 7.

### 4.1 Inter-decadal variability and trends

Although the dynamical processes which generate interannual and interdecadal variability can be studied in so-called
'control' simulations, where sources of anthropogenic climate change are neglected, comparisons of multi-decadal model
simulations with observed data become more straightforward when long-term changes of anthropogenic origin are also
reproduced by a climate model. Integrations where changes in the concentration of radiatively active gases and aerosols are
prescribed from historical data are usually referred to as 'historical' simulations. Given the simplified nature of the radiative
parametrizations in SPEEDY, it is not possible to produce historical multi-decadal simulations starting directly from
atmospheric concentration data. However, by prescribing a time-dependent absorptivity in the $CO_2$ band of the long-wave
radiation code, it is possible to simulate a change in radiative forcing leading to a realistic warming trend in the model
atmosphere. In the experiments described here, the $CO_2$ absorptivity is increased by 0.5% per year, with a prescribed
reference value being assigned for the absorptivity in 1981.

While the change in $CO_2$ absorptivity is enough to produce a fairly realistic atmospheric warming in simulation with
prescribed SST (such as those documented in Sect. 3), the coupling with the TOM3 model requires another time-varying
adjustment. This is because, when computing the Q-flux terms needed to simulate the dynamical heat transport in the ocean
(see Sect. 2.2.3), it is assumed that the annual mean tendencies of the ocean temperatures average to zero in the presence of
climatological surface heat fluxes. Since no heat flux is explicitly computed by TOM3 at the bottom of its deepest layer, the
Q term for layer 3 implicitly accounts for the heat transport between the TOM3 domain (the top 300 metres) and the deeper
part of the ocean.

On the other hand, if the annual-mean surface heat flux changes with time because of the variation in downward long-
wave radiation (as a result of the increased greenhouse effect), maintaining a constant Q-flux implies that none of the
changes in downward heat flux is transmitted to the deep ocean below the 300m depth. This would produce an overestimate
in the increase of the 300m heat content and the upper-ocean temperatures. For this reason, the Q-flux in the bottom layer of
TOM3 has to be adjusted to maintain the upper-ocean in a near-equilibrium state in the presence of long-term atmospheric
warming.





Simulations with prescribed SST running from 1950 to 2020 allow us to measure the simulated change in annual-and-global-mean surface heat-flux from the 1950s to the latest decade. Based on this estimate, and assuming a simple latitudinal variation for the bottom heat flux, the $Q_3$ is modified by adding the following term:

$$\delta Q_3 = q^*(t) \, [cos \, (\varphi)]^{\frac{1}{2}} \tag{12}$$

where $q^*(t)$ varies at an average rate of 0.8 W/m$^2$ per decade (taking into account the decreased amplitude of the latitudinal
profile at the poles, the global-mean change in heat flux is actually $\approx 0.7$ W/m$^2$ per decade).

    Since the coupled ensemble had been run with relaxation to observed SST in parts of the tropical Indo-Pacific ocean, and the sea-ice component of the model uses an observed sea-ice concentration from the ERA5 re-analysis, these two elements also contribute to the simulation of the global warming trend in our coupled ensemble.

    In Fig. 10, we show diagnostics which illustrate how well are long-term temperature variations reproduced in our coupled
ensemble. Fig. 10a shows time series of global-mean surface sea-water temperature (SSWT, excluding sea-ice temperature) anomaly with respect to the 1981-2020 average; curves are plotted for the model ensemble-mean, the individual ensemble members and the ERA5 SSWT. Apart from two short periods around 1960 and following the Pinatubo eruption in 1991 (SPEEDY does not reproduce the effects of time-varying aerosols), the SSWT ensemble-mean anomaly follows closely the ERA5 curve, the difference being mostly within the intra-ensemble variation. Interannual variations induced by the ENSO
cycle are clearly visible, confirming the role of the tropical Indo-Pacific as a pacemaker for natural variations of surface temperature (as in Kosaka and Xie 2013, 2016).

    A more interesting comparison comes from looking at the evolution of near-surface air temperature over land (Fig. 10b), taking into account that no time-varying forcing terms are included in the simple land model. Therefore, the variations of SAT over land are driven only from variations of the simulated surface heat fluxes and the advection of heat from the
oceans, with the latter process being dominant (see Compo and Sardeshmukh 2009). Again, the ensemble-mean anomaly follows closely the observed anomaly (computed from GISTEMPv4 data; Lenssen et al. 2019), with the same short periods of deviations as in the SSWT record and the evident effect of the ENSO cycle. These figures indicated that, at least on a global scale, SPEEDY is able to correctly reproduce the main factors driving interannual and interdecadal variations of surface temperature over land.

Given the very basic representation of the increase in greenhouse effect within the SPEEDY radiation code, it is instructive to verify if the latitudinal and vertical distribution of atmospheric trends are realistically simulated. The linear trend of atmospheric temperature from the SPEEDY ensemble, computed from overlapping 10-year means from 1961/70 to 2011/20, is shown in Fig. 10c. The main features seen in many observational and modelling studies are also reproduced here, including a significant polar amplification, a tropical upper-tropospheric maximum and a rather strong stratospheric cooling.
To allow a quantitative comparison with re-analysis data, the trend has also been estimated from ERA5 data taken on the same pressure levels as in the SPEEDY output; while the full cross-section is shown in Supplementary Information as Fig. S4, here we show (in Fig. 10d) a comparison of vertical profiles of area-averaged trends, namely in the tropical band (20S-




20N) and in northern high latitudes (50N-80N). For the tropical regions, the vertical profile of the warming trend in SPEEDY fits quite well the ERA5 profile, with a 50-yr variation going from about +0.5 ºK near the surface to about +1 ºK in

the upper troposphere, and then to -1 ºK in the lower stratosphere. Conversely, in the northern high latitudes the discrepancies are higher: SPEEDY shows a stronger polar amplification in the lower troposphere than ERA5, and a much stronger stratospheric cooling (≈ 3 ºK/50-yr, about three times larger than in ERA5).

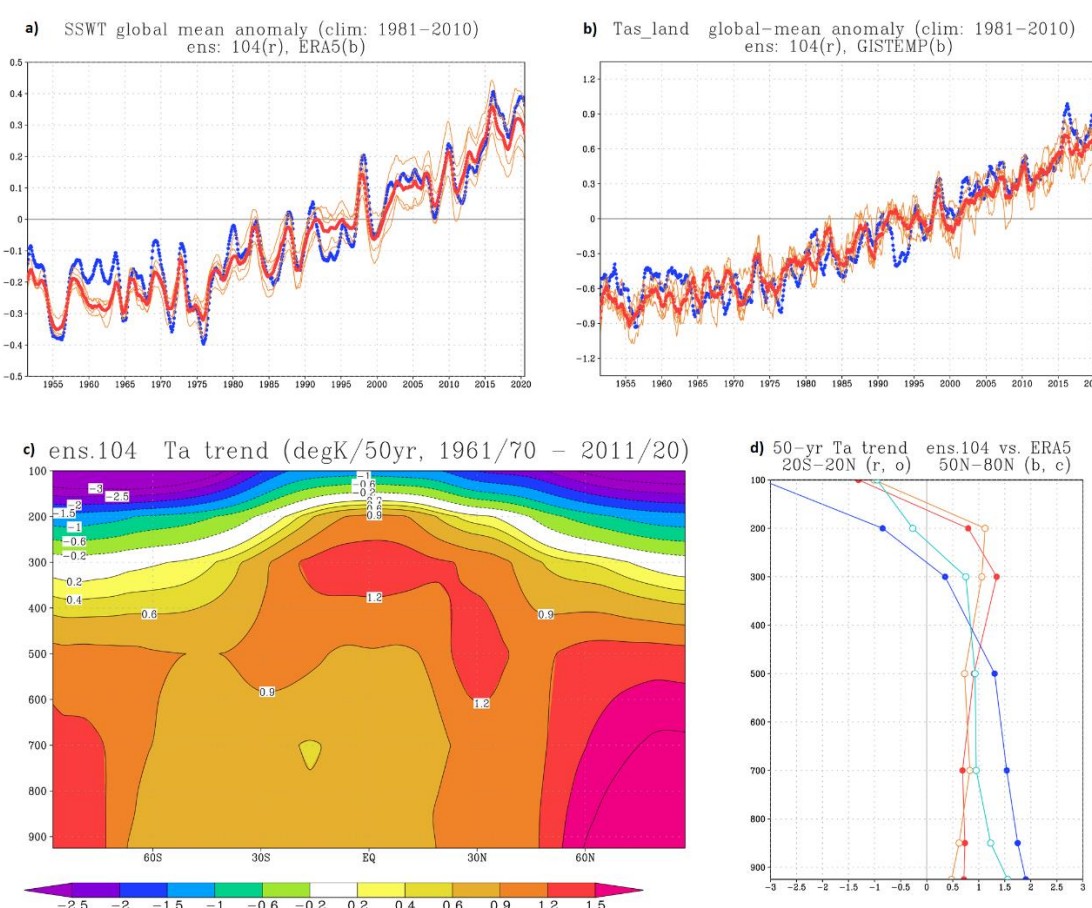


**Figure 10:** Top row: time series of global and annual-mean variability of **a)** surface sea-water temperature (SSWT), and **b)** SAT over land, from the coupled SPEEDY-TOM3 ensemble for 1951-2020 (ens.104). All data are anomalies from a 1981-2010 climatology, in ºK. Red curve: ensemble mean; orange curves: individual ensemble members; blue curve: observational data from ERA5 (for SSWT) and GISTEMPv4 (for land SAT). Bottom row: linear trends of atmospheric temperature computed from overlapping 10-yr means, from

1961/70 to 2011/20. Units: ºK/(50 yr).  **c):** vertical cross section from the coupled ensemble; **d):** vertical profiles of trends integrated in two latitudinal bands, from the ensemble in 20S-20N (red curve) and 50N-80N (blue curve), and from ERA5 in 20S-20N (orange curve) and 50N-80N (cyan curve).





Although the difference in the high-latitude stratospheric cooling trend may seem excessively large, it should be pointed out that the ERA5 trend in this region is actually much stronger (and only 10% smaller than the SPEEDY 50-yr value) during the last quarter of the 20[th] century. In the 21[st] century, observed lower-stratospheric trends are much reduced (or even reversed; see Randel et al. 2017; Philipona et al. 2018), with ozone recovery in the polar regions being a likely contributing factor. Also, the ERA5 100-hPa cooling trend is strongly asymmetrical between the two hemispheres, and is closer to the

SPEEDY value over the southern polar region (see Fig. S3).

    Given the simplicity of the SPEEDY parametrizations and time-varying forcing terms, we can conclude that the SPEEDY-TOM3 model does a reasonably good job in reproducing the surface and upper-air features of long-term warming trends. Although SPEEDY-TOM3 is clearly not an appropriate tool for detailed studies on anthropogenic climate change, it can reproduce observed SST and tropospheric trends (especially near the surface) with sufficient fidelity to allow a

meaningful validation of interannual and decadal variations even in pacemaker simulations spanning a full 70-year period.

## 4.2 Interannual variability

    We now look at how regional aspects of atmosphere-ocean variability on interannual scales are simulated in the coupled ensemble; in order to limit the influence of long-term trends on such statistics, we analyse data in the period 1981-2020.

    In Fig. 11, we show maps of SSWT standard deviation for December-February (DJF) and June-August (JJA) in the

SPEEDY-TOM3 ensemble and in ERA5. TOM3 does not reproduce interannual variations in dynamical transport, such as those associated with variability in the position of western boundary currents or in the intensity of the meridional overturning circulation; therefore we should expect the model variability to be generally lower than observed, except in regions where ocean variability is mainly driven by surface heat fluxes (and of course in the tropical areas where relaxation is activated, see Fig. 3). Overall, Fig. 11 confirms this expectation; specifically, we note the following:

●  In the tropics, SSWT variability is clearly underestimated over the Atlantic, where no relaxation is active;

    ●  Sub-tropical variability is underestimated in both hemisphere during the respective summer season;

    ●  During the boreal winter, variability in the region of the western boundary currents is also lower than observed, although by a smaller proportion than in the cases above; however, the model variability in the high-latitude sub-polar gyres is close to observations, or even larger;

●  The high-latitude North Atlantic is the region where the model variability is largest compared to observations; this can be interpreted as due to a too shallow mixed layer in TOM3 for this region, but also as an indication that changes in dynamical transport may partially compensate those induced by surface heat fluxes (as would be the case if an increase in heat transport from the tropics would follow an intensification of surface cooling and the associated convection; see Khatri et al. 2022 and references within).




**Figure 11:** Interannual variability of SSWT in DJF 1981-2020 (left) and JJA 1981-2020 (right), from the SPEEDY-TOM3 ensemble (ens.104, top) and ERA5 (bottom). Data are standard deviations of seasonal means, in ºK


Since the variability over the northern oceans in boreal winter is mostly driven by changes in surface fluxes associated with circulation anomalies, we look in more detail at the relationship between the main NH teleconnection patterns (the NAO and PNA) and the associated changes in SST over the Atlantic and Pacific oceans. In the top panels of Fig. 12, we show the covariance of 500-hPa height with the NAO and PNA indices computed from the coupled model ensemble in JF.

When these patterns are compared from those from the ensemble with prescribed SST (see Fig. 9), no significant difference



is seen for the NAO pattern, while in the case of the PNA the extension over the Atlantic Ocean is stronger in the coupled ensemble (as in ERA5).

From coupled model data, we can also compute the covariance of the teleconnection indices with SST; since the maximum ocean response is delayed with respect to the anomalies in atmospheric circulation and surface fluxes, these
covariances are computed between NAO/PNA indices in JF and SST in February-March (FM). They are shown in the second row of Fig. 12, and compared with the same diagnostics from ERA5 data in the bottom row. Overall, the TOM3 model produces SST anomalies which are highly correlated with those from ERA5. The SST anomaly induced by the PNA is also close to observations in terms of amplitude, with a partial overestimation close to the North America coast, while the SST signal associated with the NAO is stronger in TOM3 than in ERA5 over the whole North Atlantic, on average by about
50% (and almost by a factor of 2 over the Labrador Sea).

It is likely that a more accurate prescription of the mixed-layer depth in future model versions, accounting for regional variations in wind-driven mixing, may bring the amplitude of the North Atlantic signal closer to observed values. However, the high spatial correlation between the model and observed patterns is a clear indication that the TOM3 model is a suitable tool to explore the relationship between circulation, surface fluxes and temperature anomalies during the boreal cold season.
We will return to this topic in Sect. 4.3.

Finally, we investigate if the coupling to TOM3 affects the simulation of the most important tropical-extratropical teleconnection, namely the one induced by ENSO SST anomalies. Following earlier findings about sub-seasonal variations in this teleconnection (particularly over the North Atlantic; see Ayarzaguena et al. 2018 and King et al. 2021), we concentrate here on the late winter teleconnection, as represented by anomalies averaged over JF. We use the traditional
Nino3.4 SST anomaly (170W-120W, 5S-5N) as a teleconnection index, and we show in Fig. 13 the patterns of Indo-Pacific SST and NH 500 hPa height co-varying with the normalised index in the coupled and uncoupled ensembles, as well as in ERA5.

Looking first at the SST pattern in the coupled ensemble, while the positive anomalies over the central Pacific and western Indian Ocean are the obvious result of the relaxation towards observed data, the negative anomalies around the
Maritime continent and the sub-tropical Pacific appear to be realistically reproduced despite being mostly outside the relaxation domain. A second positive result is found for the extratropical teleconnection over the North Atlantic, where the dipolar pattern of height anomalies (projecting onto the negative phase of the NAO) is better simulated in the coupled than in the uncoupled ensemble: in the latter, a wider than observed negative anomaly extends over most of the North Atlantic. In both the coupled and the uncoupled ensembles, the amplitude of the North Pacific anomaly is about 25% weaker than the
observed signal, and its maximum is shifted to the east by 10 degrees approximately; while the coupled ensemble provides a better match to observations over the western North Pacific and eastern Asia, the uncoupled simulation shows a stronger amplitude over eastern Canada.





**Figure 12:** Covariances with the NAO (left) and PNA (right) indices in JF 1981-2020, as defined in Sect. 3.2. Top: covariance with 500-hPa height (in m) in the coupled ensemble (ens.104). Centre: 1-month-lag covariance with February-March SSWT (in ºK). Bottom: as in central panels, but from ERA5 data.







**Figure 13:** Covariances of SST (left, in °K) and 500-hPa height (right, in m) with the Nino3.4 SST index in JF 1981-2020. Top: from coupled ensemble (ens.104). Centre: from ensemble with prescribed SST (ens.653). Bottom: from ERA5 data. SST covariances from the prescribed-SST ensemble and ERA5 are identical by definition.




### 4.3 Relationship between sea surface heat fluxes and land surface air temperature

In this sub-section, we discuss if the SPEEDY-TOM3 model is a suitable tool to investigate the relationship between changes in the NH wintertime circulation and hemispheric-scale anomalies in land-surface air temperature, and how the circulation-induced SAT anomalies relate to long-term warming trend at regional scale (as in Molteni et al. 2017 and Yang et al. 2020). Although a thorough investigation of this topic is beyond the scope of this paper, here we assess an important prerequisite for this type of studies: namely, the ability of the model to reproduce a realistic pattern of SAT anomalies as a response to changes in planetary-scale circulation anomalies (such as the COWL pattern).

In the original definition by Wallace et al. (1996), the COWL index is computed from the difference in average lower-tropospheric temperature anomalies over the continents and the oceans in a latitudinal band covering the northern extratropics. Although the correlation between lower-tropospheric (typically below 500 hPa) and near-surface air temperature is not perfect, it is still very high during the winter season; therefore, using the original COWL definition, a positive correlation between the COWL index and average land SAT is a rather straightforward consequence.

An alternative definition, which is more indicative of the physical processes leading to extratropical SAT anomalies, can be based on the heat exchanges between the atmosphere and the land/ocean surface. This approach was adopted by Molteni et al. (2011, 2017), who aimed to relate the COWL pattern to the circulation and heating anomalies induced by thermal equilibration of planetary waves (Mitchell and Derome 1983; Marshall and So 1990). Here, we also choose to base our investigation on surface heat fluxes, but to remain closer to the original COWL definition we do not impose a specific, wave-like pattern to the heat flux anomalies (as in Molteni et al. 2011), and simply take differences between average anomalies over land and sea. Also, to focus on heat exchanges between the troposphere and the surface, we base our COWL-like index on the non-solar component of the surface heat fluxes. Specifically, we define an HF-COWL index as:

13)   HF-COWL = ave [ $F'_{ns}$ over land, 35N-70N] - ave [ $F'_{ns}$ over ocean, 35N-70N]

where $F'_{ns}$ is the anomaly of (downward) non-solar heat flux. A positive value of the HF-COWL index implies that (on average within the 35N-70N band) heat is transferred from the ocean to the atmosphere, and from the atmosphere to the land surface, at a higher rate than in climatological conditions. Note that, while SAT anomalies are generally larger over land than over the ocean, non-solar heat-flux anomalies are typically stronger over the oceans due to the contribution of latent heat flux, so there is no built-in correlation between the HF-COWL index and land SAT.

Using data from both our coupled and ensembles, and from ERA5, we have computed the HF-COWL index from seasonal DJF means from 1981 to 2020, and the covariances of this index with NH fields of non-solar heat flux, 500-hPa height and SAT. The covariance maps are compared in Fig. 14, where the standard deviation of the HF-COWL index and the average land SAT anomaly (in the 35N-70N band) are also listed above the relevant panels.

Looking first at the heat flux anomalies (left column), the model-simulated patterns show a strong spatial correlation with the ERA anomaly; the average anomaly amplitude, as measured by the HF-COWL standard deviation, is much closer to ERA5 in the uncoupled ensemble (4.3 vs 4.2 W/m$^2$ in ERA5), while is reduced by about 10% in the coupled runs (3.7



W/m$^2$). This result is to be expected, since fluxes in ERA5 are derived from short-range forecasts with prescribed SST, while in the coupled system the surface warming/cooling induced by surface fluxes acts as a negative feedback on the strength of

the downward heat fluxes.

The 500 hPa covariances shown in the middle column of Fig. 14 represent the circulation anomalies inducing (and possibly responding to) the heat flux anomaly. For both models and the re-analysis, the height anomaly shows a wavenumber-2-like pattern with lows over the northern ocean and highs at lower latitudes. However, in the ensembles the position of the ocean lows is shifted northwards, and the regions of increased ocean heat loss correspond to increased

westerly (or north-westerly) flow. Therefore, in the SPEEDY runs, heat fluxes from the ocean are enhanced not only because the air above is colder, but also because the positive westerly anomaly increases the average surface wind speed (see similar results from a seasonal forecast model in Molteni et al. 2017). The amplitude of the circulation anomaly is closer to ERA5 in the coupled ensemble than in the uncoupled one.

An even stronger advantage of the coupled model is seen in the covariances with SAT (right column in Fig. 14). The

amplitude of the SAT anomalies is much larger in the coupled than in the uncoupled ensemble, and much closer to those found in ERA5. When the SAT anomaly is averaged over land in the 35N-70N band, the SAT anomaly is close to 0.2 degrees in both the coupled ensemble and ERA5, while is only 0.05 degrees in the uncoupled experiment. Therefore, coupling to TOM3 appears to improve significantly the fidelity of the SPEEDY model in reproducing the natural contribution to SAT variability arising from increased/decreased heat transfer from the northern oceans. The coupled

simulations produce results which are comparable (at least qualitatively) to those obtained by Molteni et al. (2017) and Yang et al. (2020) from runs of with state-of-the-art coupled models, showing the potential for meaningful further investigations on this topic with the SPEEDY-TOM3 model.








**Figure 14:** Covariances of downward non-solar heat flux (left, in W/m$^2$), 500-hPa height (centre, in m) and SAT (right, in $^o$K) with the HF-COWL index in DJF 1981-2020, as defined in sect. 4.3. Top row: from coupled ensemble (ens.104). Middle row: from ensemble with prescribed SST (ens.653). Bottom row: from ERA5 data.





## 5 Discussion and conclusions

In this paper, we have described the main features of a new version (v.42) of the SPEEDY model with improved simulation
of surface fluxes, and the formulation of a 3-layer thermodynamic ocean model (TOM3) suitable to explore the coupled
extratropical response to tropical ocean variability. We have also presented results on the atmospheric model climatology,
highlighting the impact of the modifications introduced in the model code, and shown how important features of interdecadal
and interannual variability are simulated in a "pacemaker" coupled ensemble of 70-year runs, where portions of the tropical

Indo-Pacific are constrained to follow the observed variability.

   The main messages derived from our results can be summarised as follows.

- The new version of SPEEDY produces a realistic simulation of the surface heat fluxes and their annual mean balance.
    Compared to estimates from Wild et al. (2015), the main error in the model fluxes appears to be an underestimation of
    downward longwave radiation from the atmosphere; however, the global and annual mean of the net surface heat flux is

of the order of 0.5 W/m$^2$, in good agreement with current estimates and state-of-the-art model simulations.

- Changes in the parametrization of radiation and moisture diffusion have a clear positive impact on stratospheric biases
    and the Asian monsoon rainfall climatology, although they lead to a partial weakening of the NH wintertime stationary
    waves in the Atlantic sector.

- The global mean variations of SST and SAT simulated in the 70-year coupled ensemble follow observations closely,

confirming the role of the Indo-Pacific as a "pacemaker" for the natural fluctuations of global-mean surface
    temperatures (Kosaka and Xie, 2013, 2016).

- For most aspects of variability investigated in this study, the spatial patterns of anomalies simulated in the coupled
    model are highly correlated with the observed counterparts; in terms of amplitudes, regional differences may be noticed,
    such as an under-estimation of sub-tropical SST variability, a too strong polar amplification of lower-troposphere

temperature trends, and a larger-than-observed response of Atlantic SST to NAO variability.

- As in earlier versions of SPEEDY, the fidelity of the simulations (both in terms of climatological means and variability)
    is higher near the surface and in the lower troposphere, while the negative impacts of the coarse vertical resolution and
    simplified parametrizations are mostly felt in the stratosphere.

   The coupled simulations described here have been run with prescribed, time-evolving sea-ice mass. Work is now in

progress to finalise an interactive version of the sea-ice component of the model, where surface concentration and thickness
are also evolved prognostically. At the same time, better prescriptions for the mixed-layer depth and the turbulent thermal
conductivity between ocean layers are going to be tested: results shown here suggest that mixed-layer anomalies are
currently damped too strongly in regions with strong thermal stratification and too weakly in parts of the sub-polar oceans.

   In terms of scientific investigations to be performed with the SPEEDY-TOM3 model, the decadal modulation of extra-

tropical variability by tropical SST is clearly a highly suitable candidate. For example, coming back to the question raised in
Yang et al. 2020, to what extent is the decadal variability of hemispheric-scale modes such as the COWL and the Arctic



Oscillation constrained by tropical heating anomalies? Observational results suggest an influence from heat sources in the Indian Ocean and West Pacific (see Molteni et al. 2015, Jeong et al. 2022), but it is difficult to confirm such hypothesis from coupled model experiments because of a generally deficient simulation of the Indian Ocean teleconnections (Molteni et al.

2020). Although it would be too optimistic to expect SPEEDY-TOM3 to outperform many state-of-the-art models in this respect, the possibility of performing a plurality of large-ensemble experiments at a modest computational cost may overcome part of the difficulties related to the low signal-to-noise ratio associated with specific tropical teleconnections.

**Appendix: Specification of parameters in TOM3**

In this appendix, we provide further information on how some parameters controlling heat exchanges and sea-ice properties

in TOM3 are prescribed or computed.

*Turbulent thermal conductivity between ocean layers*

In Eqs. 5a and 5b, the coefficients $k_1$ and $k_2$ represent the effective thermal conductivity between the ocean layers due to turbulent and convective eddies. They are set by requiring that the heat fluxes between two layers reduce the temperature difference between the layers with a prescribed e-folding time. Considering the temperature tendency in the upper of the two

layers, we set:

$$( c_w M_1 )^{-1} F_1^w = ( T_1^w - T_2^w) / \tau_1 \tag{A1a}$$
$$( c_w M_2 )^{-1} F_2^w = ( T_2^w - T_3^w) / \tau_2 \tag{A1b}$$

and from the definition of $F_1^w$ and $F_2^w$ in Eqs. 5a and 5b (assuming an average mixed-layer depth) we derive values of $k_1$ and $k_2$, which are inversely proportional to $\tau_2$ and $\tau_2$.

For the coupled ensemble described in Sect. 4, $\tau_2$ and $\tau_2$ are set to 10 and 120 days respectively where the water temperature decreases with depth; values 5 times smaller are used where temperature increases with depth, to account for convective motions.

*Parabolic profile for sea-ice temperature*

If z is the depth measured from the ice surface, and $\theta_i$ is the ice thickness, the temperature profile within the ice is given by:

$$T^i (h) = a h^2 + b h + T_0 \tag{A2}$$

where $h = (1 - z / \theta_i )$ is an upward-oriented, non-dimensional coordinate, which is 0 at the ice lower boundary and 1 at the upper boundary. Given the mean temperature of the ice $T_m^i$, which is a prognostic variable of TOM3, first Eq. 3 is used to compute the upper-boundary temperature $T_u^i$. In Eq. 3, the parameter $\gamma_i$ varies linearly as a function of the net heat flux $F^i_{net}$ into the ice surface, with minimum and maximum values set as:

$$\gamma_i = 2 \ for \ F^i_{net} > 0 , \ \gamma_i = 2.5 \ for \ F^i_{net} < 2 F^*_{net} \tag{A3}$$



where $F^*_{net}$ is the annual-mean value of $F^i_{net}$ computed from re-analysis data over regions of total sea-ice cover. Here we assume $F^*_{net} = -12$ W/m$^2$.

The coefficients $a$ and $b$ are determined by imposing that $T^i = T_u^i$ at $h = 1$, and that the average of $T^i$ between 0 and 1 is equal to $T_m^i$. These conditions are satisfied by setting:

$$a = 3 \, ( \, T_u^i - 2 \, T_m^i + T_0 \, ) \tag{A4a}$$
$$b = 2 \, ( \, 3 \, T_m^i - T_u^i - 2 \, T_0 \, ) \tag{A4b}$$

The derivative of $T^i$ at the ice bottom, which is used to compute the heat flux at the ice-water boundary according to Eq. 5c, is given by:

$$\partial T^i / \partial z = - \, \theta_i^{-1} \; \partial T^i / \partial h = - \, b / \theta_i \quad at \; h = 0 \tag{A5}$$

For given values of $T_m^i$ and $\theta_i$, increasing the values of $\gamma_i$ in Eq. 3 decreases the amplitude of the derivative at the ice bottom and increases it at the ice surface. The limit values given in Eq. A3 ensure that $b$ is always negative, and therefore the heat flux at the ice bottom is always upward (i.e., warming the ice).

### *Partition of sea-ice heat content tendency into temperature and mass changes*

As discussed in Sect. 2.2.3, TOM3 assumes that the time derivative of sea-ice heat content, computed from the net heat flux
into the ice, is converted partly into a temperature change and partly into an ice mass change (see Eqs. 7b and 8). We assume that the proportion of heat converted into a mass change decreases with increasing ice mass, because with increasing thickness the water-ice boundary becomes gradually more insulated from the changes occurring at the ice surface. Therefore, the fraction of heat $\alpha_i$ converted into a temperature tendency (Eq. 7b) must increase with ice mass; specifically, we assume:

$$\alpha_i = 0.5 \, [ \, 1 + min \, ( \, 1, \, 2 \, f_i^2 \, )] \tag{A6}$$

For small ice fractions the change of heat content is partitioned equally between the temperature and mass contributions, while the ice mass cannot grow beyond a mass fraction $f_i = (0.5)^{1/2}$.

### *Relationship between sea-ice concentration and thickness*

Sea-ice mass can be either prescribed or evolved prognostically in TOM3. In the first mode, time-evolving fields of sea-ice surface concentrations are available from different observational sources (here, we use ERA5 data), but ice thickness data are
available only for limited periods. In the second mode, the model predicts the fraction of ice mass $f_i$, and a diagnostic relationship between surface concentration and thickness must be specified to compute these two variables from the ice mass.

In order to ensure a consistency between the two operating modes, in both cases we assume that, for concentration values $s_i < 1$, the ice thickness $\theta_i$ varies linearly between a prescribed minimum value $\theta_1$ and a maximum value $\theta_2$:

$$\theta_i = \theta_1 + s_i \, ( \, \theta_2 - \theta_1 \, ) \tag{A7a}$$



The equivalent equation for the water-equivalent depth (as a fraction of the layer-1 depth) is:

$$d_i = d_1 + s_i ( d_2 - d_1 ) \tag{A7b}$$

where values of $\theta_i$ and $d_i$ are related as in Eq. 2b. For the experiments described in this paper, $\theta_1 = 0.6$ m and $\theta_2 = 1.5$ m.
In the interactive mode 2, Eqs. 2a and A7b are combined into a quadratic equation relating $f_i$ and $s_i$ for $f_i < d_2$:

$$f_i = s_i [ d_1 + s_i ( d_2 - d_1 ) ] \tag{A8}$$

Once $s_i$ is computed from Eq. A8, $d_i$ is set to $f_i / s_i$. If $f_i \geq d_2$, we set $s_i = 1$ and $d_i = f_i$.

In mode 1 (used in our pacemaker ensemble), the ice thickness is prescribed from Eq. A7a if $s_i < 1$. If $s_i = 1$, a provisional thickness value $\theta_i'$ is estimated from the climatological surface ice temperature $T_u{}^i$ (from ERA5), assuming that the vertical temperature gradient is equal to an annual-mean gradient $\beta_i$ computed from re-analysis data in regions of full ice-cover:

$$\theta_i' = ( T_0 - T_u{}^i ) / \beta_i \tag{A9}$$

where we set the temperature at the ice bottom to be equal to the freezing temperature $T_0$ of sea water. If $\theta_i' \leq d_2$, we set $\theta_i = d_2$; otherwise:

$$\theta_i = \theta_2 + w_i{}^* ( \theta_i' - \theta_2 ) \tag{A10}$$

where the weight $w_i{}^*$ is a function of the climatological concentration averaged over the six winter/spring months of each
hemisphere (December to May for the NH). In this way, when ice cover is 100% at a given time, the prescribed thickness is determined by the climatological surface-to-bottom temperature difference in regions where full ice cover persists throughout the cold season.

The climatological mean thickness obtained from Eqs. A7a, A9 and A10 for the periods February-March and August-September 1981-2010 is shown in supplementary Fig. S5 for the northern polar regions. Although these estimates may differ
substantially from observed thickness data, they should be considered as an improvement with respect to the assumption of a fixed thickness, and provide a consistent approach (at least for $s_i < 1$) between the two operating modes of the TOM3 ice model.



*Code and data availability.* The model code used for this paper is available upon request from the corresponding author.

The model output for the prescribed SST and coupled ensembles are available on Zenodo respectively as:

Ens. 653: Molteni et al. 2023a,  https://doi.org/10.5281/zenodo.7947858

Ens. 104: Molteni et al. 2023b,  https://doi.org/10.5281/zenodo.7947932

Observational datasets used in the verifications are available from the following web sites:

ERA5 re-analysis: https://climate.copernicus.eu/climate-reanalysis

GPCP precipitation: https://psl.noaa.gov/data/gridded/data.gpcp.html

GISTEMPv4 surface air temperature: https://data.giss.nasa.gov/gistemp/

*Authors contributions.* The development of the SPEEDY model is the results of a long-standing collaboration among the authors; the TOM3 model was designed by FM and developed in consultation with the other authors. The ensemble experiments were run by FM, and all authors contributed to the assessment of the results.

*Competing interests.* The authors declare no competing interests.

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
