# Peer review of "Multi-decadal pacemaker simulations with an intermediate-complexity climate model"

_EGUsphere, 2023_

## Author Comment (AC1)

**Reply to 'RC1: Comment on egusphere-2023-1103' by Referee 1 (Noel Keenlyside)**

We are grateful to Dr. Keenlyside for his many constructive comments. Our reply is as follows.

**Major comments**

➢ The idea of prognostic equations for sea ice thickness is interesting (mode 2), but I found the discussion of mode 1 (prescribed sea ice thickness) and mode 2 confusing. As I understand, mode 2 has not been tested. Given this and that you don't use mode 2 in the paper, it does not seem appropriate to introduce the prognostic computation of sea ice thickness in this paper.

We agree that the prognostic equation for ice mass is not strictly relevant to this paper since the described experiments are all run with prescribed ice concentration and thickness. However, the experiments use the prognostic equation for ice temperature, which (through the introduction of the parameter $\alpha_i$ ) is formulated in such a way to be consistent with the implementation of interactive ice mass in mode 2. We believe that showing (in Eq. 8) how the results of Eqs. 7a and 7b are used in the interactive-ice mode makes clear why the equations are formulated in that way. However, we acknowledge that the original text was not completely clear about what is actually applied in mode 1, so we have modified it as follows:

For sea ice, the time derivative of the heat content is given by:

$$\partial HC_l^i / \partial t = s_i ( F_s^i + F_{ns}^i ) - s_i F_0^i \tag{7a},$$

However, the model assumes that only a fraction $\alpha_i$ of the heat entering the ice layer is converted in temperature change, since in the real world part of energy gain/loss is used to decrease/increase the ice mass. Specifically, we assume that ice mass increases when sea ice is cooling and decreases when sea ice is warming; therefore the ice temperature tendency is reduced, in absolute value, with respect to schemes which assume a constant ice thickness. With this assumption, which for consistency applies to both modes of the sea-ice scheme, the prognostic equation for sea ice mean temperature is given by:

$$( c_i \ f_i \ M_l ) \ \partial T_m^i / \partial t = \alpha_i \partial HC_l^i / \partial t \tag{7b}$$

Starting from values of $HC_l^i$ and $T_m^i$ at time t, Eqs. 7a and 7b are used to compute the corresponding values at time t+δt. When the sea-ice scheme is run in operating mode 1, only the change in ice temperature is retained, while the ice mass at each time step is derived from prescribed, time-evolving values of ice concentration and thickness. If instead the ice scheme is run in mode 2 (i.e. with interactive mass), the mass fraction of sea ice $f_i$ at time $t+\delta t$ is set to the value which satisfies the heat content conservation:

$$HC_l^i (t+\delta t ) = M_l f_i (t+\delta t) [ c_i (T_m^i (t+\delta t ) - T_0) - L_f] \tag{8}$$

and an empirical relationship is used to compute values of $s_i$ and $d_i$ at time $t+\delta t$ from the updated value of $f_i$ (see Appendix).

➢ In section 4 you begin with a simulation of SPEEDY-TOM3 in pacemaker mode. However, there was no discussion about the model's climate and how it compares to the prescribed SST runs described in section 3. Has coupling with TOM3 altered the model climatology? Also, to what extent does the model's reproduction of global warming relate to the pacing of Indo-Pacific SST (and prescribed sea ice). The performance of the SPEEDY-TOM3 configuration without pacing the SST and prescribed sea ice should be described, as this would be a more natural control experiment (which prospective model users would need to perform).

In order to reply to this comment, we discuss some results for an ensemble of simulations performed without tropical SST relaxation, which we used to verify if the Q-flux terms were appropriate to maintain a realistic SST

climatology. The introductory part of Sect. 4 has been considerably expanded to include this discussion, and two new figures (included below) have been added to the Supplementary Information: Fig. S4 comparing biases of SST and 500-hPa height with and without relaxation, and Fig. S5 showing global warming trends from the ensemble without relaxation. The revised text at the start of Sect. 4 is as follows:

We now present results from multidecadal simulations run with SPEEDY v.42 coupled to the TOM3 model. Two 5-member ensembles were run; both ensembles used prescribed ice concentration and thickness, while the temperature of sea water and sea ice evolved interactively by TOM3 according to Eqs. 6 and 7 in Sect. 2.2. In order to ensure consistency with the time-evolving radiative forcing used in multi-decadal SPEEDY simulations (see section 4.1 below), both ensembles used prescribed ice concentration and thickness derived from ERA5 data.

- A 5-member coupled ensemble (v.42c) was run for the period 1980-2020 without any relaxation to observed SST. The 5 members were initialised from slightly different atmospheric initial conditions, while the initial conditions for the water temperature in the TOM3 model were set, for all members, to the values from the WOA09 climatology. The main purpose of this ensemble is to verify that the Q-flux term in the TOM3 equations maintains an SST climatology close to the observed one for the recent decades (the Q-flux terms were actually computed using data for 1981-2010). Due to the absence of tropical ocean dynamics (and the associated teleconnections), the interannual variability in this ensemble cannot be compared with observed data.

- A second 5-member ensemble has been run from January 1950 to December 2020 with relaxation to observed SST activated in parts of the tropical Indo-Pacific ocean (as explained in Sect. 2.4); this is referred to as the pacemaker ensemble (v.42p). As in v.42c, perturbed initial conditions were used for the atmospheric fields, while in all members TOM3 was initialised from climatological fields appropriate for the decade started in 1950. These were obtained as a modification of the WOA09 climatology, adding corrections proportional (at any grid point) to the difference between ERA5 SST in the 1950-1959 and the 1981-2010 periods; an appropriate scaling factor and a seasonal time lag were defined for the second and third TOM3 layer based on observational data.

A comparison between the SST bias in v.42c and v.42p are shown in Supplementary Fig. S4 for winter and summer seasons in 1981-2010. In both ensembles, the bias is mostly negative (apart from the polar regions) and is largest over the extratropical oceans in the summer hemisphere, with a typical amplitude of $0.4 \sim 0.5$ °K. Smaller positive biases are found over the North Atlantic and North pacific during boreal winter. By construction, the SST bias of the pacemaker v.42p ensemble is null over the relaxation domain in the Indo-Pacific, but it is also reduced over the tropical Atlantic, as a result of atmospheric teleconnections linking the different tropical basins. On the other hand, the difference in tropical SST bias has no discernible impact on the extratropical atmospheric circulation. The bias of 500-hPa height during the boreal winter, also shown in Fig. S4, is practically identical in the two ensembles, and shows no significant difference from the bias of the prescribed SST ensemble shown in Fig. 8c.

In the following sub-sections, the interannual and decadal variability simulated in the pacemaker ensemble will be presented and compared with the observed variability; as stated above, no meaningful comparison with observed interannual variability can be done on v.42c, due to the absence of tropical coupled dynamics (ENSO *in primis*) and the associated teleconnections, while global warming trends are discussed in the next sub-section.

See also our reply to comments on L531-533 and L583-585 below.

**Detailed comments.**

➢ *L21 "observed statistics" -> "that observed"*
Corrected

➢ *L76-80, it is not entirely clear why this information is included in the introduction. Is it an important new feature of the model?*
➢ *L110-111, why is there an interpolation to a finer horizontal grid?*
We have found that computing the surface fluxes on a higher-resolution grid produces a more accurate simulation of surface fluxes in regions of strong SST gradients and near the coastlines, especially during the winter season, when the differences in surface temperature between land and sea is very large. Although the fluxes computed from the coarser atmospheric grid would be suitable for the atmospheric simulation (after the contributions from the land and sea fraction in each grid box are averaged), the ocean fluxes may be unrealistically strong, leading to potential problems in a thermodynamic ocean model.

➢ *L230, I assume that you must then add a corresponding term to the third layer (equal and opposite to the fluxes in the upper two layers) to account for the convective mixing also.*
$F_1^w$ and $F_2^w$ are fluxes defined at the interfaces between the model layers, so any change to these fluxes produces compensating effects in the layers above and below the interface (see Eq. 6a-c) and the total heat content is conserved.

➢ *L261, Have you used a value of 1.5m for ice thickness in this paper? Please make this clear.*
We have removed the reference to the thickness value used in ERA5, which was confusing; we now specify that in ice mode 1 we use "prescribed, time-evolving values of ice concentration and thickness" (see reply to main comment 1). The actual thickness values are shown in Supplementary Fig. S7.

➢ *L322, Fluxes from AGCM simulations are known to be inconsistent with SST in regions where the SST variations are driven by turbulent fluxes, as in such AGCM experiments the SST are not influenced by the fluxes. Thus, it is not clear to me why you prefer to use an AGCM experiment for estimating fluxes.*
We think it is more appropriate to compare the SPEEDY ensemble with prescribed SST with a state-of-the-art ensemble using the same kind of boundary conditions (the SST used to drive both ensembles are derived from HadISST2, which is also used in ERA5). If fluxes from a coupled ensemble had been used in the comparison, differences in surface fluxes may arise from SST biases.

➢ *L333-338, This could be written a little more clearly, by indicating the difference for the global average solar fluxes, and then writing "global average of the net surface heat flux" (rather than "net global averages").*
➢ *In figures 4 and 5, it would be useful to also show differences maps. I can see differences between the ERA and SPEEDY patterns, but the text glosses over them.*
Difference maps between SPEEDY and ECMWF fluxes are now shown in a new version of Supplementary figure S2, included below (which also shows differences from the earlier SPEEDY version, as requested by Rev. 2). The text commenting these figures is now as follows:

The spatial patterns of solar and non-solar fluxes from SPEEDY, shown in Fig. 4, compare quite well with the ECMWF counterparts. The SPEEDY net solar radiation is lower than the ECMWF flux over most of the tropical oceans, higher in the extratropics and over the tropical continents (see Fig. S2). The SPEEDY global average is about 6 W/m$^2$ larger than the ECMWF value, and just outside the range estimated by Wild et al. (2015), but still quite realistic. A positive difference of about 6 W/m$^2$ with respect to the ECMWF model is also found in the global average of the (upward) non-solar heat flux, resulting in a close match between the global averages of the net surface heat flux: 0.5 W/m$^2$ for SPEEDY and 0.7 W/m$^2$ for the ECMWF model, both values being well within the range estimated by Wild et al. (2015). Also, the maps for the net surface radiation in the two models (bottom

panels in Fig. 4) show a good correspondence between ocean regions gaining or losing energy, an important requisite if users want to couple SPEEDY to an ocean model.

When looking at the individual components of the non-solar heat flux (in Fig. 5), the net longwave radiation emerges as the fields with the larger discrepancies between SPEEDY and the ECMWF model, with the former showing larger values except over the tropical Indo-Pacific ocean. …

➢ *It would be useful to show biases for rainfall shown in figure 7.*
Unfortunately the rainfall bias maps are quite noisy, and affected by the difference between the real (high and steep) topography of South Asia and the smooth model topography. We believe that the improved simulation of monsoon rainfall in version 42 is already clearly visible from the total rainfall maps in Fig. 7, and in view of the large number of multi-panel figures already included in the paper we prefer to keep this figure as it is.

➢ *It is confusing to refer to "ens. 653" in the caption and the figure titles.*
We have changed the captions using (hopefully) more intuitive acronyms to indicate the various ensembles.

➢ *L498, it is not possible to understand the initial conditions for the TOM3 from this sentence.*
More information has been added, see text in the second bullet in the reply to Major comment 2 above.

➢ *L531-533, it would be useful to see a simulation without such changes. How does the model behave without prescribing changes in tropical SST?*
➢ *L583-585, It should be explained that SPEEDY-TOM3 as described here has an imposed tropical SST warming. This may explain the large agreement with observations.*
In response to these comments, and to similar ones from Rev.2, we have added two panels in Fig. 10 (below), showing trends of SST and SAT from the pacemaker ensemble only in the extratropical regions not affected by observational constraints. We have also added a new Supplementary Figure (Fig. S5) showing the global trends from the ensemble without SST relaxation. Text added to Sect. 4.1 is as follows:

It should be remarked that, in our pacemaker ensemble, SSWT over most of the Indo-Pacific Ocean and the sea-ice concentration are constrained to follow the ERA5 values. It is therefore appropriate to look at the variability of SSWT and SAT averaged only in the extratropical regions not affected by such constraints. Time series of SSWT and SAT averaged in the extratropical domain (25N-65N and 25S-65S) are shown in Fig. 10c and 10d respectively, using the same format as in the panels for global values. The removal of the tropical domain clearly reduces the signature of ENSO-driven interannual variability in the SSWT time series, but the overall trends are still closely captured over both sea and land. Consistently, a realistic simulation of SSWT and SAT trends was also found in the v.42c ensemble, where SSWT relaxation is not applied (see Supplementary Fig. S5).

➢ *L616-617, Is the increased signal of the PNA in the North Atlantic Ocean an indication that O-A is important for this teleconnection? Or does it reflect some differences in the model climatology between the SST and TOM3 experiments?*
Based on the 500-hPa height biases shown in Fig. 8 and Supplementary Fig. S4, we added this sentence:

As discussed at the beginning of Sect.4, the bias in 500-hPa height climatology is very close between the uncoupled and coupled ensembles, therefore the stronger PNA extension cannot be attributed to a different atmospheric climatology in the coupled runs.

➢ *L686, I think you mean "coupled and prescribed SST ensemble simulations". I am wondering whether the analysis from the models is computed using the ensemble means, and how this might affect the comparison with the observations (which is one realisation).*
The quoted text has been corrected. We now specify in Sect. 3.2 that all covariances are computed from individual members' data, not from ensemble means. Also, in Sect. 4.3, the new text reads:

Using data from both our pacemaker and uncoupled ensemble members, and from ERA5, we have computed the HF-COWL index …..

The information has been added to the caption.

➢ L698, do you mean "both coupled and uncoupled ensemble simulations"?

Corrected

Corrected

Corrected

Corrected

➢ L801, I don't follow why you say the ice mass cannot grow beyond an ice fraction. I think you mean the maximum rate occurs when fi=0.5^0.5; It is also confusing that you discuss thickness above, but actually you use ice fraction in A6.1.

The revised text is as follows:

We assume that the proportion of heat converted into a mass change decreases with increasing ice mass, because with increasing thickness (which in TOM3 is a monotonic function of ice mass, see below) the ice at the lower boundary becomes gradually more insulated from the changes occurring at the ice surface. Therefore, the fraction of heat $\alpha_i$ converted into a temperature tendency (Eq. 7b) must increase with ice mass; specifically, we assume:

$$\alpha_i = 0.5 \left[ 1 + min ( 1, 2 f_i^2 )\right] \tag{A6}$$

For small ice fractions the change of heat content is partitioned equally between the temperature and mass contributions, while for $f_i = (0.5)^{1/2}$ the change of heat content is entirely converted into a temperature change, and therefore ice mass cannot grow beyond that value.

[Figure]

**Figure 10:** Top row: time series of global and annual-mean variability of **a)** surface sea-water temperature (SSWT), and **b)** SAT over land, from the SPEEDY-TOM3 pacemaker ensemble for 1951-2020 (v.42p). All data are anomalies from a 1981-2010 climatology, in ºK. Red curve: ensemble mean; orange curves: individual ensemble members; blue curve: observational data from ERA5 (for SSWT) and GISTEMPv4 (for land SAT). Middle row: **c)** and **d)** as in a) and b) respectively, but only for the extratropical domain (25N-65N, 25S-65S) where no observational constraint is applied. Bottom row: linear trends of atmospheric temperature computed from overlapping 10-yr means, from 1961/70 to 2011/20. Units: ºK/(50 yr).  **e):** vertical cross section from the pacemaker ensemble (v.42p); **f):** vertical profiles of trends integrated in two latitudinal bands, from the ensemble in 20S-20N (red curve) and 50N-80N (blue curve), and from ERA5 in 20S-20N (orange curve) and 50N-80N (cyan curve).

[Figure]

**Figure S2:** Difference between annual-mean surface heat fluxes from SPEEDY and ECMWF ensembles with prescribed SST. Left: SPEEDY v.42 minus ECMWF historical ensemble (Roberts et al. 2018); right: SPEEDY v.42 minus SPEEDY v.41. Top: net solar radiation; centre: net longwave radiation; bottom: turbulent (sensible + latent) heat flux. Global-mean values are listed above each panel. Unit: W/m$^2$.

[Figure]

**Figure S4:** Average biases of SSWT in December-February (top, in °K), SSWT in June-August (centre, in °K), 500 hPa height in December-February (bottom, in dam) with respect to ERA5 data in years 1981 to 2010, for the SPEEDY coupled ensemble without SST relaxation (v.42c, left) and the pacemaker ensemble (v.42p, right).

[Figure]

**Figure S5:** Time series of global and annual-mean variability of surface sea-water temperature (SSWT, left), and SAT over land (right), from a SPEEDY-TOM3 coupled ensemble for 1980-2020 without relaxation to observed tropical SST (v.42c). All data are anomalies from a 1981-2010 climatology, in °K. Red curve: ensemble mean; orange curves: individual ensemble members; blue curve: observational data from ERA5 (for SSWT) and GISTEMPv4 (for land SAT).

---

## Author Comment (AC2)

**Reply to 'RC2: Comment on egusphere-2023-1103' by Referee 2**

We are grateful to the referee for his positive and constructive comments. Our reply is as follows.

**Major comments:**

➢ Section 3.1: A comparison with the heat fluxes of the previous version of the model would be useful in the main text (similar to section 3.2). Have the heat fluxes improved in the newer version due to the modifications?

To address this topic, we have added global-mean values of fluxes derived from a prescribed-SST ensemble run with the previous model version (v.41), and we have plotted differences between surface fluxes from the two version in the revised Supplementary Fig. S2 (included below). The following text has been added in Sect. 3.1:

When comparing the surface heat fluxes in SPEEDY v.42 with those produced by the previous model version (again in an ensemble with prescribed SST), the main improvements are the reductions of the net solar radiation and sensible heat flux, while in v.41 the global averages of net longwave radiation and latent heat flux were slightly closer to the values by Wild et al. (2015). Overall, the net global balance is much improved in v.42, decreasing from 3.6 W/m$^2$ to 0.5 W/m$^2$. Spatial maps of the differences between the fluxes in the two model versions are shown in the right-hand column of Fig. S2.

|  | SPEEDY v.42 | SPEEDY v.41 | ECMWF-Ah | Wild et al. (2015) |
|---|---|---|---|---|
| Net solar radiation (downw.) | 167.8 | 175.0 | 162.1 | 160 (154/166) |
| Net longwave radiation (upw.) | 69.8 | 67.3 | 58.0 | 56 (48 / 62) |
| Sensible heat flux (upw.) | 17.3 | 22.2 | 17.6 | 21 (15 / 25) |
| Latent heat flux (upw.) | 80.3 | 81.9 | 85.8 | 82 (70 / 85) |
| Net surface heat flux (downw.) | 0.48 | 3.57 | 0.74 | 0.6 (0.2 / 1.0) |

**Table 1** Global averages of surface heat fluxes (for the 1981-2010 period) in multidecadal ensemble simulations with SPEEDY v.42, SPEEDY v.41 and the ECMWF atmospheric model (ECMWF-Ah; Roberts et al. 2018), all with prescribed SST, compared with mean estimates and uncertainty ranges from Wild et al. (2015). All data in W/m$^2$.

➢ Section 4.1: Figure 10 a and b: I think it would be useful to have similar Figures for the NH extratropical SSWT and extratropical land TAS anomaly. Wondering if the good agreement for the global means are dominated by the tropics - large parts of the tropical SSWT are prescribed.

In response to this comment, and to a similar one from Ref.1, we have added two panels in Fig. 10 (below), showing trends of SST and SAT from the pacemaker ensemble only in the extratropical regions not affected by observational constraints. We have also added a new Supplementary Figure (Fig. S5) showing the global trends from a coupled ensemble without SST relaxation (v.42c). Text added to Sect. 4.1 is as follows:

It should be remarked that, in our pacemaker ensemble, SSWT over most of the Indo-Pacific Ocean and the sea-ice concentration are constrained to follow the ERA5 values. It is therefore appropriate to look at the variability of SSWT and SAT averaged only in the extratropical regions not affected by such constraints. Time series of SSWT and SAT averaged in the extratropical domain (25N-65N and 25S-65S) are shown in Fig. 10c and 10d respectively, using the same format as in the panels for global values. The removal of the tropical domain clearly reduces the signature of ENSO-driven interannual variability in the SSWT time series, but the overall trends are still closely captured over both sea and land. Consistently, a realistic simulation of SSWT and SAT trends was also found in the v.42c ensemble, where SSWT relaxation is not applied (see Supplementary Fig. S5).

➢ Section 4.3: Figure 14 and corresponding text: My impression is that there are substantial differences between the coupled model and ERA. The coupled model patterns have a structure that reminds me of the NAO, while the ERA patterns remind me more of the East Atlantic pattern. Therefore, the physical processes involved in the COWL-pattern might be different in the model and in the reanalysis data - in particular the importance of a dynamic ocean cannot be ruled out.

The referee is correct in pointing out these differences, although it is difficult to attribute them to a specific cause. We speculate that in ERA5 meridional advection of polar air over the Western North Atlantic may play a bigger role, as stated in the modified text of Sect. 4.4:

… However, in the pacemaker ensemble the position of the ocean lows is shifted north-westwards, in a NAO-like pattern, and the regions of increased ocean heat loss correspond to increased westerly (or north-westerly) flow. Therefore, in the SPEEDY coupled runs, heat fluxes from the ocean are enhanced not only because the air above is colder, but also because the positive westerly anomaly increases the average surface wind speed (see similar results from a seasonal forecast model in Molteni et al. 2017). In ERA5, the increase in surface heat fluxes on the western North Atlantic is likely to be due to increased meridional advection of polar air on the western side of the negative geopotential anomaly.

**Minor comments:**

➢ *line 401 fi g 7b => Fig. 7b*
corrected

➢ *line 686 Using data from both our coupled and ensembles => Using data from both our coupled and uncoupled ensembles?*
Corrected

[Figure]

**Figure 10:** Top row: time series of global and annual-mean variability of **a)** surface sea-water temperature (SSWT), and **b)** SAT over land, from the SPEEDY-TOM3 pacemaker ensemble for 1951-2020 (v.42p). All data are anomalies from a 1981-2010 climatology, in ºK. Red curve: ensemble mean; orange curves: individual ensemble members; blue curve: observational data from ERA5 (for SSWT) and GISTEMPv4 (for land SAT). Middle row: **c)** and **d)** as in a) and b) respectively, but only for the extratropical domain (25N-65N, 25S-65S) where no observational constraint is applied. Bottom row: linear trends of atmospheric temperature computed from overlapping 10-yr means, from 1961/70 to 2011/20. Units: ºK/(50 yr). **e):** vertical cross section from the pacemaker ensemble (v.42p); **f):** vertical profiles of trends integrated in two latitudinal bands, from the ensemble in 20S-20N (red curve) and 50N-80N (blue curve), and from ERA5 in 20S-20N (orange curve) and 50N-80N (cyan curve).

[Figure]

**Figure S2:** Difference between annual-mean surface heat fluxes from SPEEDY and ECMWF ensembles with prescribed SST. Left: SPEEDY v.42 minus ECMWF historical ensemble (Roberts et al. 2018); right: SPEEDY v.42 minus SPEEDY v.41. Top: net solar radiation; centre: net longwave radiation; bottom: turbulent (sensible + latent) heat flux. Global-mean values are listed above each panel. Unit: W/m$^2$.

[Figure]

**Figure S5:** Time series of global and annual-mean variability of surface sea-water temperature (SSWT, left), and SAT over land (right), from a SPEEDY-TOM3 coupled ensemble for 1980-2020 without relaxation to observed tropical SST (v.42c). All data are anomalies from a 1981-2010 climatology, in $^{o}$K. Red curve: ensemble mean; orange curves: individual ensemble members; blue curve: observational data from ERA5 (for SSWT) and GISTEMPv4 (for land SAT).

---

## Author Response (AR2)

**Reply to 'RC1: Comment on egusphere-2023-1103' by Referee 1 (Noel Keenlyside)**

We are grateful to Dr. Keenlyside for his further comments on the revised manuscript. Our reply is as follows.

**Major comments**

> *The authors have addressed all but one of my concerns. I still find the description of how sea ice is treated in mode 1 and mode 2 confusing. I think the authors can be easily address this issue with a little rewriting. Below is a summary of my understanding and points of confusion.*
> *After reading through the relevant sections several times, I have understood the following for mode 1: The fraction of ice mass is computed using Eq 2a, 2b, A7a from the prescribed sea ice concentration. However, at line 264, it also states that thickness is also prescribed. Thus, the exact setting for mode 1 are ambiguous.*
> *The sea ice temperature is computed using equations 7a, 7b, and A6, using the fraction of ice mass. The introduction of an adjustment for sea ice temperature that accounts for the formation/melting of ice is interesting. However, the values in equations A6 are A7a could be a little better justified.*
> *I am sorry but it is still not clear to me why including equations A7b – A10 is necessary, and how it helps to explain the formulation in equation A6. Furthermore, it is unclear why you use Si<1 and Si=1 rather than Si<=1. Doesn't the formulation introduce a discontinuity? And as far as I could tell the values of d1 and d2 are not given. The description of the prognostic formulation for the fraction of sea ice mass, then the Appendix "Relationship between sea-ice concentration and thickness" could benefit from reordering to first discuss mode 1, and then discuss mode 2.*

We believe that the confusion originated from the fact that, in the Appendix section *"Relationship between sea-ice concentration and thickness"* aspects common to mode 1 and mode 2 were discussed first, and then specific aspects of the two operating modes were described. We admit that this approach made it less clear to understand what parts actually applied to the experiments described in the paper.

Therefore, we have rewritten this section following the Reviewer's suggestion, discussing first the formulation applied in Mode 1 (lines 847-872, Eqs. A7 to A9), and then describing the Mode 2 relationship in the last few lines of the Appendix (lines 874-879, Eq. A10). We have also specified the values of $d_1$ and $d_2$ in Eq. A7b as requested, and the value of β in Eq. A8. With regard to Eq. A6, we now specify in lines 844-845:

The parameters in Eq. A6 have been empirically chosen is such a way to achieve a realistic annual cycle of sea-ice temperature.

In the main text, at line 264, we have now spelled out clearly how we prescribe sea-ice concentration and thickness in Mode 1, making references to the relevant equations in the Appendix. The new text (lines 263-270) is as follows:

When the sea-ice scheme is run in operating mode 1, only the change in ice temperature is retained, while the ice mass at each time step is derived ==from time-evolving values of ice concentration (prescribed from ERA5 data) and thickness (estimated from ERA5 ice concentration and surface temperature, see Eqs. A7 to A9 in the Appendix).== If instead the ice scheme is run in mode 2 (i.e. with interactive mass), the mass fraction of sea ice $f_i$ at time $t+\delta t$ is set to the value which satisfies the heat content conservation:

$$HC_1{}^i\,(t+\delta t\,) = M_1\,f_i\,(t+\delta t)\,[\,c_i\,(T_m{}^i\,(t+\delta t\,) - T_0\,) - L_f\,] \tag{8}$$

and an empirical relationship is used to compute values of $s_i$ and $d_i$ at time $t+\delta t$ from the updated value of $f_i$ ==(see Eq. A10 in the Appendix).==

**Minor comments**

➤ L34, *several is an understatement. Haven't there been "numerous" studies with SPEEDY.*
"several" changed to "numerous" as suggested.

➤ L64, *fullstop missing.*
Corrected

**Reply to 'RC2: Comment on egusphere-2023-1103' by Referee 2**

We are grateful to the referee for their further comments on the revised manuscript. Our reply is as follows.

**Minor comments:**

➤ *line 64: dot missing before 'Although'*
Corrected

➤ *- line 386: atmospheric temperature in the lower troposphere (at 100 hPa) => atmospheric temperature in the lower stratosphere?????? (at 100 hPa)*
"lower troposphere" corrected into "lower stratosphere"